# Magnetite Nanoparticles: Synthesis and Applications in Optics and Nanophotonics

**DOI:** 10.3390/ma15072601

**Published:** 2022-04-01

**Authors:** Nataliia Dudchenko, Shweta Pawar, Ilana Perelshtein, Dror Fixler

**Affiliations:** 1Department of Chemistry, Bar-Ilan Institute of Nanotechnology & Advanced Materials (BINA), Bar Ilan University, Ramat Gan 5290002, Israel; natalia.dudchenko@biu.ac.il (N.D.); ilana.perelshtein@biu.ac.il (I.P.); 2Bar-Ilan Institute of Nanotechnology & Advanced Materials (BINA), Faculty of Engineering, Bar Ilan University, Ramat Gan 5290002, Israel; sppawar.shweta@gmail.com

**Keywords:** magnetite nanoparticles, magnetic ferrofluids, synthesis, application, optical devices

## Abstract

Magnetite nanoparticles with different surface coverages are of great interest for many applications due to their intrinsic magnetic properties, nanometer size, and definite surface morphology. Magnetite nanoparticles are widely used for different medical-biological applications while their usage in optics is not as widespread. In recent years, nanomagnetite suspensions, so-called magnetic ferrofluids, are applied in optics due to their magneto-optical properties. This review gives an overview of nanomagnetite synthesis and its properties. In addition, the preparation and application of magnetic nanofluids in optics, nanophotonics, and magnetic imaging are described.

## 1. Introduction

Nanotechnology is a field of fundamental and applied science dealing with methods of research, methods of production, and the use of materials with defined atomic structures through the controlled manipulation of individual atoms and molecules. Nanotechnology is focused on the creation, investigation, and application of new types of materials, called “nanomaterials”, of which the size of at least one dimension is from 1 to 100 nm. Nanomaterials find their application in the broadest fields, including the chemical industry [1,2,3], agriculture [4,5], engineering [6], pharmaceutics [7], sustainable energy [8], medicine [9,10,11], etc.

In the last two decades, nanomaterials such as magnetic nanoparticles (MNPs) were widely utilized for a huge number of applications due to their specific magnetic properties, high surface area, and unique dimensions. Among them are NPs of different magnetic structures: nanodots [12], nanorods [13], nanowires [14,15], nanotubes [16], core–shell nanoparticles [17], etc. The chemical composition of MNPs also varies widely, including pure metals, alloys, metal oxides, and doped NPs. MNPs could be further coated and functionalized according to the needs of their consumers. Here, we highlight magnetite nanoparticles among a giant family of MNPs.

Considerable efforts are devoted to the creation of synthetic MNPs and to various strategies for their synthesis. In general, two main approaches to nanomagnetite synthesis exist: top-down and bottom-up. Top-down approaches consist of breaking up large pieces of the initial material to produce nanoparticles and include the use of physical methods, such as ball milling, electron beam lithography, etc. Bottom-up approaches imply the creation of nanoparticles from molecules and include the use of both chemical and biological methods. Attaining NPs in the range of nanometers using physical methods is difficult; thus, more efforts should be directed toward creating and improving chemical and biological methods of synthesis in order to obtain MNPs with desirable sizes, shapes, crystallinities, and compositions. However, a question remains to be answered: what method is most suitable for MNP production? Every method has its limitation, so the choice of synthesis method depends on the desired application of the MNP.

As MNPs are easily oxidizable by oxygen in the environment and agglomerate due to their large specific surface area, developing specific functional coatings (surface functionalization) that would protect the MNPs and further decorate MNPs with therapeutic agents, fluorescent labels, etc. is necessary.

MNPs are thought to be promising for a wide range of applications because of their unique characteristics, including their high saturation magnetization, which makes them easily operated by the magnetic field, and their low toxicity. They are broadly applied and have potential for biomedical applications, such as magnetic drug targeting [18,19,20], DNA/RNA purification [21,22,23], magnetofection [24,25], hyperthermia [26,27,28,29], MRI imaging [30,31], cell separation [32], etc. MNPs also could be utilized for the creation of ferrofluids (FF)—colloidal suspension of NPs in polar or non-polar liquid carriers. The main distinctive features of FFs are the fluidities of liquid material and the magnetism of MNPs. FFs have unique magnetic properties in the presence of magnetic fields such as optical anisotropy, birefringence, and field-dependent transmission. These unique properties of FFs make them a promising platform for novel optoelectronic devices, such as optical fiber sensors, optical gratings, organic light-emitting diods, agents for the enhancing of MRI images, photonic materials, etc. The use of FFs in optics and nanophotonics faces numerous challenges; thus, investigations in this field are necessary.

This review gives a comprehensive overview of nanomagnetite properties and synthesis as well as focuses on recent developments in MNPs for optical application. First, the structure and special properties of magnetite are briefly discussed (Section 2). We also discuss various methods of nanomagnetite synthesis as well as the preparation of magnetic FFs (Section 3 and Section 4). Then, the application of magnetic FFs in optics and nanophotonics is highlighted (Section 5). Finally, we discuss some challenges in this field of research and their possible solutions (Conclusions). This review not only promotes recent state-of-the-art methods of magnetite synthesis (biological routes) in comparison with routine chemical methods but also discusses novel cutting-edge applications of magnetite nanoparticles (magnetite ferrofluids) in optics and nanophotonics, such as “smart” windows, solar energy harvesting, organic light-emitting diodes, etc.

## 2. Structure and Properties of Magnetite

Magnetite, Fe_3_O_4_, an iron oxide, has an inverse spinel structure (Figure 1). The details of its structure were first established in 1915 by W.H. Bragg [33], who provided one of the first mineral structures determined using the X-ray diffraction method. Magnetite has a cubic unit cell with a lattice constant of *a* = 0.839 nm and is an iron oxide of mixed valences Fe^3+^ and Fe^2+^ within the stoichiometry Fe^2+^/Fe^3+^ = 0.5. The main feature of the spinel crystal lattice is the presence of two crystallographic positions of iron ions: Fe^2+^ and half of the Fe^3+^ ions occupying octahedral sites (surrounded by six oxygen O^2−^), and the other half of the Fe^3+^ ions occupying tetrahedral sites (surrounded by four oxygen O^2−^). Magnetite is frequently non-stoichiometric, resulting in a cation-deficient Fe^3+^ layer. Fe^2+^ could also be fully or partly replaced by other divalent ions (Mg^2+^, Co^2+^, Ni^2+^, etc.), which leads to changes in the lattice constant.

The magnetic properties of magnetite are determined by its crystal structure. Namely, the ordering of magnetic spin moments of iron cations in the crystal lattice occurs through a special interaction between electrons of the 3d shell of neighboring cations. The ordering of magnetic spin moments leads to the overall magnetic ordering in magnetite. Magnetite possesses magnetocrystalline anisotropy, a special case of magnetic anisotropy that is related to spin-orbit coupling. Magnetocrystalline anisotropy depends on the magnetite crystal structure, shape, and chemical composition and plays a critical role in the magnetic ordering of magnetite. The unique magnetic and structural properties of magnetite, in comparison, for example, with ferrites, are determined also by the presence of so-called hopping electrons. Briefly, 3d electron transfer occurs in the Fe^2+^–Fe^+3^ octahedral cation pair, which could be described in short as Fe^2+^(3d^6^) ↔ Fe^3+^(3d^5^). The concentration of hopping electrons in magnetite is high in comparison with spinel ferrites, which have similar lattice structures, and thus makes a significant contribution to the formation of its magnetic properties. To conclude, magnetite at room temperature is ferrimagnetic with a Curie temperature of 580 °C (Table 1), which has a rather high saturation magnetization value of 92 A·m^2^/kg.

## 3. Synthesis and Characterization of Magnetite Nanoparticles

MNPs for various applications should possess certain parameters, namely, narrow size distribution, biocompatibility, weak aggregation (stability), definite surface properties, and high saturation magnetization. To fulfill the abovementioned requirements, a wide diversity of synthetic methods was developed. Here, we highlight two routes for magnetite nanoparticle preparation: chemical and biological routes.

### 3.1. Chemical Synthesis of Magnetite Nanoparticles

Among diverse techniques developed for nanomagnetite synthesis, we could mention the co-precipitation method, partial oxidation of ferrous hydroxide, reactions in constrained environments, hydrothermal or high-temperature reactions, polyol method, sol–gel reactions, sonochemical procedure, etc. Many authors [35,36,37,38,39] reported the features of this chemical technique and indicated their advantages and disadvantages.

#### 3.1.1. Co-Precipitation Method

The most commonly used method is the co-precipitation method first reported by Massart [40], which has been extensively investigated and modified. This synthesis leads to the formation of magnetite nanoparticles of defined size and high magnetic properties. Magnetite nanoparticles are obtained by co-precipitation of ferrous and ferric salts in the water medium at different Fe^+3^/Fe^2+^ ratios, with a strong basic solution (i.e., ammonia) at room or elevated temperatures [41,42]. The reaction between the iron salts solution and the precipitating agent is slightly exothermic [43]. The principal scheme of the magnetite synthesis via the co-precipitation method is represented in Figure 2.

The following reactions were proposed for the mechanism of magnetite formation [45]. At the first stage, ferric and ferrous hydroxides are precipitated in the water solution:Fe^3+^ + 3OH^−^ = Fe(OH)_3_(1)
Fe^2+^ + 2OH^−^ = Fe(OH)_2_(2)

At the second stage, ferric hydroxide decomposes to FeOOH:Fe(OH)_3_ = FeOOH + H_2_O,(3)

Finally, a solid state reaction between FeOOH and Fe(OH)_2_ takes place, resulting in magnetite formation:2FeOOH + Fe(OH)_2_ = Fe_3_O_4_ + 2H_2_O,(4)

The overall reaction is as follows:2Fe^3+^ + Fe^2+^ + 8OH^−^ = 2Fe(OH)_3_Fe(OH)_2_ → Fe_3_O_4_ + 4H_2_O,(5)

The concentration and size of magnetite nanoparticles are influenced by various factors, including initial concentrations and molar ratios of ferrous and ferric salts, base solution concentrations and nature, reaction temperature, presence/absence of external magnetic field and/or microwave irradiation, inert atmosphere, stirring rate, etc.

A Scanning Electron Microscopy (SEM) image of magnetite nanoparticles synthesized by the co-precipitation method is shown in Figure 3.

This method is simple and enables a large number of nanoparticles to be obtained. We note that, among the disadvantages of the method, controlling the shape of the nanoparticles is difficult (poor morphology), and particles with impurities (non-stoichiometric magnetite) are obtained.

#### 3.1.2. Partial Oxidation of Ferrous Hydroxide

The next widely used procedure for nanomagnetite fabrication was first reported by Sugimoto and Matijevic [46]. They reported the next route of monodispersed spherical nanomagnetite synthesis. Amorphous ferrous hydroxide was first precipitated from the ferrous sulphate solution, and then the aqueous gel was aged at 90 °C in the presence of nitrate ion, which results in magnetite formation. This route yields “monodispersed” spherical magnetite particles over a broad range of diameters.

The properties of MNPs obtained by this method depend on the concentration of the reagents, the pH of the reaction mixture, the nature of the nonconstituent anions and cations, the presence/absence of oxygen in the aging system, and the influence of an external magnetic field [47].

This method is convenient and has some advantages: the versatility of the method and the fact that surface of these particles does not need to be modified further to make them hydrophilic. The latter is especially relevant for biomedical applications of magnetic iron oxide nanoparticles [48]. The disadvantages of the method are the formation of goethite in the presence of minor amounts of oxygen and extensive agglomeration of the synthesized NPs [49].

#### 3.1.3. Reaction in Constrained Environments

One of the ways to produce magnetite nanoparticles with uniform dimensions is to perform the reaction in constrained environments. Different synthetic and biological reactors could be used for this synthesis, such as apoferritin protein cages [50], phospholipid membranes (micelles) [51], mesoporous materials [52], microemulsions [53], etc. Here, we would focus on apoferritin protein cages as constrained environments for magnetite nanoparticles synthesis.

In 1992, Mann was the first who synthesized magnetite inside a horse spleen ferritin cavity [54]. Ferritin is an iron storage protein and has a spherical protein cage with an exterior diameter of 12 nm and an interior cavity diameter of 6–8 nm. Yu et al. [50] have a reported novel thermostable ferritin (PcFn), purified and characterized, that could successfully direct the synthesis of thermostable magnetoferritins (M-PcFn) with monodispersed iron oxide nanoparticles in one step. They obtained good crystalline and superparamagnetic magnetite nanoparticles with an average diameter of 4.7 nm (Figure 4). Xue et al. [55] gave a comprehensive analysis of the synthesis, properties, modifications, and biomedical applications of magnetite nanoparticles, synthesized inside a ferritin cavity. Uchida et al. [56] describes a protocol of nanoparticle synthesis in protein cages and the synthesis of magnetite (or maghemite) nanoparticles (hard inorganic materials) within ferritin cages.

The cage-like property of ferritin is an ideal template for the synthesis of various nanoparticles as the protein shell completely isolated the newly synthesized nanoparticles, preventing their oxidation and particle interaction. It is a unique biomineralization system that makes the final biomineral soluble and mobile yet biochemically inert [57]. Nowadays, magnetite nanoparticles, synthesized inside ferritin protein shells, are widely used for biomedical applications such as MRI contrast agents, cell targeting, gene therapy, etc.

#### 3.1.4. Hydrothermal or High-Temperature Reactions

Hydrothermal or high-temperature reactions are another wet-chemical method of synthesizing magnetite nanoparticles performed in a reactor or autoclave in an aqueous media, where a pressure of >6000 Pa and a temperature of >200 °C are maintained [39].

To date, magnetite nanoparticles of various morphologies, such as spheres, cubes, and wires have been synthesized using this method. The morphology of synthesized nanoparticles could be determined by changing the reaction temperature, solvent type, precursor salt, reducing agent, etc. Torres-Gómez et al. [58] presented a simple and efficient method for pure phase magnetite nanoparticle synthesis via the hydrothermal method. The authors tuned the morphology of obtained nanoparticles by changing the reaction temperature; the nanoparticles were synthesized at 120 °C, 140 °C, and 160 °C. All of the nanoparticles obtained were shown to be Fe_3_O_4_ in a pure magnetite phase with a high level of crystallinity and uniform morphology at each temperature (Figure 5). The synthesized nanoparticles exhibited good saturation magnetization, and the resulting shapes were quasi-spheres, octahedrons, and cubes. Cursaru et al. [59] reported the physical-chemical, structural, magnetic, and biocompatible properties of magnetite prepared using the hydrothermal method in different temperature and pressure conditions. The authors found that crystallite size increases with increases in pressure and temperature while the hydrodynamic diameter is influenced only by temperature. The magnetic core of particles synthesized at high temperatures is larger (13 nm) than the magnetic cores of the particles, synthesized at 100 °C (11 nm). The particles show only minor toxicity, meaning that these particles could be suited for biomedical application. Kumar et al. [60] presented a detailed investigation of the temperature effect on the hydrothermal synthesis of magnetite nanoparticles. They showed that temperature plays a crucial role in the single-phase synthesis of magnetite nanoparticles. Thus, a single-phase cubic magnetite structure with high crystallinity was found in the samples synthesized at 160 and 180 °C, whereas samples prepared at 120 and 140 °C were of a mixed phase (α-Fe_2_O_3_ and Fe_3_O_4_).

We can conclude that by using a hydrothermal or high-temperature reaction, one could obtain magnetite nanoparticles of different shapes and morphologies. Among the advantages of this method, we mention the high crystallinity of the nanoparticles obtained. The main disadvantage of this method is the need for high temperatures and pressures, which means expensive facilities.

#### 3.1.5. Polyol Method

The polyol method is usually utilized to obtain magnetite nanoparticles with controlled shapes and sizes. The type of polyol, salt ratio, salt concentration, etc. affects the growth, shape, size, and yield of the particles [39]. Polyols are both reducing and stabilizing agents used to control particle growth, which also prevents aggregation of the particles.

Wan et al. [61] declared a simple and effective route for obtaining a new class of magnetite nanoparticles based on a high-temperature decomposition of ferric acetylacetonate in triethylene glycol at elevated temperatures. The synthesized nanoparticles were uniform in size, highly crystalline, and superparamagnetic at room temperature. The unique hydrophilic surface structures of the particles were shown to lead to stability not only in aqueous solutions at a neutral pH but also in a physiological buffer. The authors claimed that these novel magnetite nanoparticles should have great potential as high-performance MRI contrast agents for cell or molecular imaging and diagnostic applications. Oha et al. [62] synthesized superparamagnetic magnetite nanoparticles using the polyol method, which is based on the polar polyol. They found that the particle size decreased as the concentration of sodium acetate increased and decreased when the molar ratio of the iron precursor was decreased. They obtained magnetite particles ranging from 11 nm to 338 nm in size. Hachani et al. [63] synthesized iron oxide nanoparticles of low polydispersity through a polyol synthesis in high-pressure and high-temperature conditions. The authors claimed that this process yields nanoparticles with a narrow particle size distribution (Figure 6) in a simple, reproducible, and cost-effective manner without the need for an inert atmosphere. Their potential as a magnetic resonance imaging (MRI) contrast agent was confirmed.

To conclude, the advantages of polyol synthesis are its low cost and hydrolytic stability. Among the disadvantages of this method are its thermal instability and flammability.

#### 3.1.6. Sol–Gel Synthesis

The sol–gel method is a wet-chemical process for nanoparticle preparation based on hydrolysis and polycondensation of iron precursors with the formation of a colloidal solution of nanoparticles (“sol”) and further drying (“gel” formation) to remove the solvent and, finally, to obtain magnetite nanoparticles [64].

Shaker et al. [65] reported about magnetite nanoparticles prepared using the sol–gel method combined with annealing at temperatures of 200, 300, and 400 °C. The characterization results showed that the size of magnetite nanoparticles can be changed by varying the annealing temperature. They claimed that the sol–gel method offers several advantages for the preparation of magnetite nanoparticles: the synthetic process is economical and environmentally friendly, and size-controlled magnetite nanoparticles are produced at different annealing temperatures. Takai et al. [66] demonstrated the preparation of magnetite nanoparticles using a sol–gel-assisted method with further annealing in a vacuum at different temperatures (200–400 °C). The results indicated that magnetite nanoparticles of different sizes were obtained, simply by varying the annealing temperatures. The morphologies of the particles obtained at 400 °C are more spherical. The higher temperature, the higher the mean particle size of the nanoparticles obtained.

The advantages of sol–gel synthesis include its monodispersity, good control of the particle size, control of the microstructure, desirable shapes and lengths of the products, its high purity, and good crystallinity of the products. Among the disadvantages of this method are its long completion time, toxic organic solvents, and contamination of the product with the matrix component.

#### 3.1.7. Sonochemical Synthesis

Via the sonochemical procedure, the chemical reaction occurs due to the application of ultrasound irradiation, which causes acoustic cavitation in the aqueous solutions. Such cavitation leads to the formation, growth, and collapse of bubbles that produce a great amount of energy and create high temperatures and pressures. This method was also suitable for the synthesis of magnetite nanoparticles.

De Freitasa et al. [67] demonstrated that the preparation of MNPs using the sonochemically assisted homogeneous co-precipitation method resulted in nanoparticles with sizes of ~35 nm. The authors concluded that the smaller diameter nanoparticles were obtained by applying a higher power and demonstrated iron oxide nanoparticle synthesis at a pH lower than 6 for the first time. The conclusion was made that ultrasonic energy accelerates the reaction. Wang et al. [68] synthesized magnetite nanoparticles of a controlled size (~15 nm) and size distribution via an ultrasound-assisted co-precipitation technique. The nanoparticles obtained have rather high saturation magnetization (~50 A·m^2^/kg). Nevertheless, the authors concluded that the magnetic characteristics of the nanoparticles obtained were lower than that of nanoparticles produced without ultrasound assistance due to the size difference. Fuentes-García and co-authors [69] developed a fast, single-step sonochemical strategy for the green manufacturing of magnetite nanoparticles, using iron sulfate as the sole source of iron and sodium hydroxide as the reducing agent in an aqueous medium. They reported that the sodium hydroxide concentration was varied to optimize the final size and magnetic properties of the magnetite nanoparticles and to minimize the number of corrosive byproducts from the reaction. The proposed method allows for control of the uniformity in size and shape of the produced MNPs within the ≈20−60 nm range, making them promising for future biomedical applications.

The advantages of the sonochemical procedure are that the method reduces the growth of crystals and increases the reaction rate. One could obtain nanoparticles with high crystallinity, saturation magnetization, and narrow size distribution. However, the mechanism of this action is still not well understood, so the shape and size of the nanoparticles obtained are difficult to control.

### 3.2. Biological Synthesis of Magnetite Nanoparticles

MNPs also could be produced using biological objects such as bacteria, plants, fungi, etc. Biological synthesis is an efficient, environmentally friendly, and green process [39]. Because of this, the particles produced are less stable, have less homogeneity, and agglomerate more [70]. The principal scheme of green synthesis for MNPs is shown in Figure 7.

One of the methods of MNP production consists of mixing precursor salts with a green substrate containing biological compounds [71]. Green substrates act as both reducing and limiting agents, which can stabilize the NPs during the synthesis process. To obtain NPs with different properties, one could modify the concentrations of precursor salt and green substrate, time, temperature, and pH parameters during the synthesis. The authors claimed that this approach is not only simple but also economical and produces less waste, which makes it environmentally friendly. The MNPs synthesized using this method are suitable for biomedical applications due to a coating formed of the biological compounds of the green substrate that is not toxic and biocompatible [38].

Magnetotactic bacteria, which was first discovered by Blakemore in 1975 [72], also could be utilized for MNPs’ synthesis. This process is called biologically controlled mineralization and results in the formation of iron oxide NPs covered by a lipid bilayer membrane, called magnetosome. Nowadays, some cultured strains of magnetotactic bacteria could be used for MNP synthesis. The chemical composition, shape (morphology), and crystal size of MNPs are uniform for each strain. Magnetite crystals in magnetosomes of cultured strains could be in the form of a prism, a cuboctahedron, a bullet, or a combination of a cube and dodecahedron [73]. Biologically synthesized MNPs possess features that differ from chemically obtained nanoparticles. They have a narrow size distribution, a uniform morphology, and high levels of magnetite crystallinity (Figure 8) [74]. They are usually single magnetic domain NPs, which means that their magnetic moment is thermally stable at a physiological temperature. The arrangement of magnetosome chains inside the bacteria prevents aggregation, thus yielding a high rate of internalization within human cells. Magnetosomes are covered with lipids that result in a negative charge of their surface; thus, they can easily be functionalized. They exhibit high biocompatibility and low toxicity towards living organisms. All of the abovementioned advantages make magnetotactic bacteria a promising substrate for the creation of magnetite nanoparticles.

Although biological synthesis of MNPs demonstrates eco-friendliness and non-toxicity, regarding chemical synthesis, it also experiences great challenges, namely, low yield of NPs as well as the need for further investigation of the mechanisms behind its synthesis.

To conclude Section 3, we have difficulty recommending one method as the most suitable for MNP production. Both chemical and biological routes have their own advantages and disadvantages as well as challenges that need to be overcome (Table 2).

As every method has its limitation, the choice of synthesis method needs to be made based on the needs of the desired application of MNPs.

## 4. Preparation of Magnetic Ferrofluids

The first synthesis of ferrofluid was reported by Pappel [75] in 1965, and after that, the number of publications in this area increased each year. Nanoparticles of ferromagnetic metals as well as magnetic compounds can be used to produce magnetic ferrofluids (FFs). The stability of FFs is one of the most critical challenges, and achieving appropriate stability in FFs remains a major challenge. Second, the increase in viscosity caused by the use of nanofluids is a significant disadvantage due to the increased pumping power required.

The two main steps involved in the synthesis of magnetic ferrofluids are (i) the production of nano-sized MNPs and (ii) the subsequent dispersion of the nanoparticles obtained into carrier liquids [76]. Two types of magnetic fluids can be synthesized depending on the method of colloidal stabilization: water-based fluid or organic surfaced-based fluids. Typically, the FFs consist of a magnetic phase, a carrier liquid, and additives. Nanosized iron oxide is one of the most widespread magnetic phases in ferrofluids. Various approaches were developed to synthesize and characterize high-quality magnetic iron oxide nanoparticles. These nanoparticles are classified by the phases in which they form: liquid phase, gas phase, and solid phase (Figure 9).

MNPs are synthesized in the liquid phase by acoustic-chemical, microwave [77], thermal decomposition [78], and chemical reduction methods [79] as well as by co-deposition and microemulsion [80]. Chemical gas-phase deposition, arc discharge [81], and laser pyrolysis are used to produce them in the gas phase, while combustion and annealing are used to obtain them in the solid phase. An alternative method is grinding a magnetic powder of micron sizes for several weeks in a ball mill by mixing it with a solvent and dispersant [82].

In general, two approaches are used in producing nanofluids: one-step and two-step. The manufacture of nanoparticles and the preparation of ferrofluid are coupled in the one-step technique. The nanoparticles are manufactured initially and then dispersed in a suitable carrier fluid in the second stage of the two-step procedure. Magnetic powders are suspended in a carrier liquid, which acts as a medium [83].

The protection of colloid nanoparticles from oxidation and the prevention of agglomeration and coagulation in both the manufacturing and transformation of particles into the colloidal state in the carrier liquid are significant goals in the production of FFs. These fluids are created in a variety of carrier liquids, including water, silicone oil, synthetic or semi-synthetic oil, mineral oil, lubricating oil, kerosene, and mixtures of these and many other polar liquids. The carrier liquid must be inert with both the magnetic phase and the substances used in the device. As an alternative to the traditional alcohol- and water-based FFs, other solvents such as ionic liquids were tested. Studies on FFs employing ionic liquids were published in recent years, and they appear to be a promising subject of research. Rodriguez-Arco and colleagues [84] described the synthesis of magnetic fluids made up of magnetite nanoparticles suspended in ionic liquid in the presence of different additives. To ensure the long-term stability of ionic liquid-based FFs, steric repulsion was required, and the easiest approach to accomplishing this was to use surfactants adsorbed on the surface of the particles with tails compatible with the liquid carrier [84]. Dispersants must be used during the FFs synthesis to reduce particle agglomeration and to promote colloidal stability, both of which are significant in FF applications. Nanofluids must have features such as durability and stability, as well as a low proclivity for agglomeration. The additives must also be chosen to fit the carrier liquid’s dielectric characteristics. Oleic acid (OA) is a typical surfactant used to stabilize MNPs produced using the classic co-precipitation method [83]. Citric and tartaric acids, in addition to oleic acid, are utilized to stabilize FFs within a wide range of pH (pH 3–11) [85]. Polymers such as silica [86], chitosan [87], polyvinyl alcohol (PVA) [88], and ethylene glycol [89] are commonly employed to coat nanoparticles and to prevent them from sticking together, thus improving aqueous medium dispersibility. To avoid oxidation, anti-oxidant compounds might be added. pH control additives are also employed in water-based FFs. The use of ultrasonication or the addition of surfactants can successfully reduce agglomeration and can improve nanoparticle dispersion behavior. A schematic representation of the water-based magnetic ferrofluids preparation is proposed in Figure 10.

The use of oil blend-based Ferro-nanofluids to improve thermal conductivity for heat transfer applications was reported by Imran et al. [91]. This paper showed how to make oil-based Ferro-nanofluids more efficiently with tunable thermal conductivity for heat transfer applications. Elsaidy et al. [92] used size-customized clusters of iron oxide nanoparticles to tune the thermal characteristics of aqueous nanofluids. Mohapatra and co-workers [93] synthesized ferrofluid oil droplets containing superparamagnetic nanoparticles coated with a surfactant or polymer distributed in water to form magnetic nanoemulsion. This work showed how pH and magnetic field control may be used to tune the optical characteristics of magnetic fluids.

To conclude, the stability of ferrofluids is critical for their different applications, which have been increasing in recent years due to their tunable properties.

## 5. Application of Magnetic Ferrofluids in Optics and Nanophotonics

Magnetic fluids are utilized for various technological, biological, and medical purposes, including machine element design applications (magnetic sealing, interior and viscous dampers, FF film bearing, and FF lubrication), bio-medical applications (site-specific drug delivery for hyperthermia and contrast agents for MRI), and thermal engineering applications (enhancement of thermal conductivity of FFs and pool boiling heat transfer of FFs) [94,95]. In recent years, magnetic FFs found application in optics due to their unique magneto-optical properties in the presence of the magnetic field, such as optical anisotropy, birefringence, and field-dependent transmission.

### 5.1. Photonic Materials

Photonic materials are prospective materials for different technologies, such as tunable photonic crystals, switches, filters, as well as tags for biological applications, and the methods of their production now are developing rapidly. Ge et al. [96] developed a method for the preparation of polyacrylate-capped superparamagnetic magnetite colloidal nanocrystal clusters with tunable sizes from 30 to 180 nm. After the synthesis of MNPs by high-temperature hydrolysis of ferric iron with the addition of sodium hydroxide in a solution containing polyacrylic acid and diethylene glycol, the NPs obtained were dispersed in water and remained stable in solution for at least several months. Such superparamagnetic clusters were reported to be able to be employed directly in the construction of colloidal photonic crystals with highly tunable stop bands that can be moved across the entire visible spectral region owing to the highly charged polyacrylate-capped surfaces and the strong interaction of the magnetite colloidal nanocrystal clusters with a magnetic field. The tuning range of the diffraction wavelength is related to the average size of the colloidal nanocrystal clusters. In general, large clusters (160–180 nm) preferentially diffract red light (~750 nm) at relatively weak (~87 G) magnetic fields, while small clusters (60–100 nm) diffracted blue (~430 nm) light in stronger (~350 G) magnetic fields. The optical response of the photonic crystals to external magnetic was concluded to be rapid and fully reversible, which could provide a new platform for the fabrication of novel optical microelectromechanical systems, sensors, and color display units.

Yang et al. [97] reported the gram-scale hydrothermal synthesis of superparamagnetic magnetite nanocrystal clusters and their long-term charge stability. Their investigation of the yield and particle size of the products with different reaction volumes, from 20 mL to 340 mL, revealed that the hydrothermal route is scalable for preparing magnetite nanocrystal clusters at the gram scale or even larger. The assembly and photonic properties of the as-prepared magnetite nanocrystal clusters dispersed in water in response to external magnetic fields were studied. For the colloidal nanoclusters prepared at sizes of ~123 nm, a diffraction peak at 780 nm appeared when a magnetic field of 50 Oe was applied and the diffraction peak gradually blue-shifted from 780 to 470 nm as the magnetic field strength increased. The applications of the magnetite nanocrystal clusters as building blocks for aqueous-form magnetically responsive photonic crystals with widely, rapidly, and reversibly tunable diffractions across the visible and near infrared ranges as well as long-term stability of photonic performances were also demonstrated.

Kostopoulou et al. [98] developed a modified synthetic protocol for the growth of monodispersed, superparamagnetic, colloidal nanoclusters, which consisted of iron oxide nanocrystals with adjustable sizes. The colloidal nanoclusters’ physical behaviors were studied, and the influence of the grain size on their magnetic and optical response was highlighted. The optical properties of the nanoclusters created were shown to be able to be tuned by applying an external magnetic field: with an increase in the strength of the external magnetic field, in the range of 160–600 G, the diffraction shifted towards the blue region of the spectrum. The nanoclusters were concluded to combine useful attributes, such as superparamagnetic behavior, with high saturation magnetization; to be stable in water; and to be able to be easily synthesized from commercial starting materials.

He et al. [99] developed magnetically tunable photonic structures by assembling superparamagnetic magnetite colloidal nanocrystal clusters with overall diameters in the range of 100–200 nm. They showed that instantly (less than 1 s) assembling clusters into ordered structures and rapidly tuning their photonic properties across the whole visible region through the application of a relatively weak (typically 50–500 Oe) external magnetic field are possible.

Wang et al. [100] constructed full-color tunable photonic materials using superparamagnetic citrate-capped magnetite nanoparticles with a spherical shape and an almost uniform particle diameter. The average diameter of the magnetite nanoparticles was ~135 nm, and the saturation magnetization was ~70 A·m^2^/kg. Superparamagnetic nanoparticles were assembled as photonic materials in deionized water through the application of a magnetic field. In the absence of an external magnetic field, a classical FF appeared in the superparamagnetic colloids, and the structural color of the colloids appeared black. However, when a magnetic field was applied to the magnetite colloids, the structural color of colloids immediately changed (Figure 11).

Thus, under an external magnetic field, an orderly photonic structure forms. Magnetite colloids exhibit tunable structural colors under a magnetic field ranging from 20 to 800 G, and their reflection spectra have a wavelength range of approximately 420–800 nm, which includes the entire visible range. Therefore, the nanoparticle diameter, nanoparticle concentration, and shell thickness of the core–shell nanoparticles affect the reflection intensity and spectral range. The abovementioned properties of synthesized nanoparticles are promising candidates for achieving full-color display, and they also help in understanding the mechanism of photonic materials.

The process of photonic material preparation is shown in Figure 12.

To conclude, a wide range of magnetite nanoparticles was used for the fabrication of photonic materials. Superparamagnetic magnetite nanocrystals or nanoclusters with tunable sizes from 30 to 200 nm, long-term stability, and rather high saturation magnetization could be directly utilized for the constructing of colloidal photonic crystals through the application of a magnetic field. The optical properties of such nanoclusters can be tuned by applying an external magnetic field (typically 50–500 Oe), and the reflection spectra of synthesized superparamagnetic colloids have a wavelength range of approximately 420–800 nm, which includes the entire visible range.

### 5.2. Organic Light-Emitting Diodes (OLEDs)

Organic light-emitting diodes (OLEDs) exhibit unique properties such as light brightness and color tunability and are prospective candidates for full-color displays and solid-state lighting applications. They are usually fabricated using organic molecules or conjugated polymers. During recent years, the role of nanoparticles of different types became significant for OLED fabrication [101] due to their unique optical, electronic, and magnetic properties. Magnetite nanoparticles were one of the metal oxide nanoparticles that are also utilized for the improvement of OLED properties.

Zhang et al. [102] constructed a new approach to realizing the light extraction enhancement in OLEDs using gold-coated magnetite nanoparticles with an optimized diameter. The time-resolved photoluminescence and optical haze characterizations were performed to study the influence of synthesized nanoparticles on the lifetime of excitons and the light-scattering effect, respectively. The results reveal that the localized surface plasmonic effects induced by gold nanoparticles and the strong light-scattering effect related to synthesized gold-doped magnetite nanoparticles lead to double enhancement in light extraction, which paves a new path for realizing high-efficiency and controllable OLEDs.

Lian et al. [103] analyzed the effect of three hybrid composite hole-injection layers, namely, magnetite nanoparticles that are surface modified with graphene, silicon dioxide, and gold nanoclusters. By optimizing the concentration of MNPs in hole-injection layers, an obvious improvement in the performance of MNP-contained OLEDs was realized compared with the control device. The enhancement in the electroluminescence efficiency of the OLEDs was attributed to the combination of light-scattering, localized surface plasmon resonance, and magnetism, leading to a simultaneous increase in the internal quantum efficiency and the out-coupling efficiency. The authors anticipated that the method developed will have great potential application in high-performance OLED displays and solid-state lighting.

Organic light-emitting diodes (OLEDs) are usually fabricated using organic molecules or conjugated polymers. During recent years, magnetite nanoparticles were also utilized for improvements in OLED properties. Gold-doped magnetite nanoparticles were demonstrated to efficiently enhance the luminescence of OLEDs due to the combination of light-scattering, plasmon resonance (gold), and magnetism (magnetite). Such a combination is very promising for the fabrication of new, highly efficient OLEDs.

### 5.3. Magnetic Field Sensors

Magnetic field sensors integrated with magnetic FFs have attracted considerable attention because of their ease of fabrication, high sensitivity, and low cost [104] as well as due to their unique magneto-optical properties [105]. Cennamo et al. [106] presented a novel methodology for magnetic field sensing by exploiting surface plasmon resonance (SPR) sensors based on D-shaped plastic optical fiber (POF) and magnetic fluids. The proposed sensor system utilizes a ferrofluid layer, deposited on a patch as input to a multimode SPR-POF sensor, to change the mode profile inside a SPR-POF platform. A proof-of-concept for this novel sensing approach was obtained by exploiting a prototype sensor that was characterized in the range between 0.15 and 1.2 mT. In the linear range of the sensor response, sensitivity was estimated to be about 6800 pm/mT, and the resolution was estimated to be about 0.029 mT. Moreover, the comparison between the expected and experimental behaviors showed a very good match with a mean squared error of about 5%.

Zheng et al. [107] proposed and demonstrated an intensity-modulated magnetic field sensor operated in reflection mode based on a magnetic fluid-coated tilted-fiber Bragg grating (TFBG) cascaded by a chirped fiber Bragg grating (CFBG). Transmission of the TFBG is modulated by the refractive index of the magnetic fluid, which is sensitive to an external magnetic field. The magnetic fluid used in this experiment was commercially available water-based FF fabricated using the chemical co-precipitation technique. It is a black-brown translucent liquid with a refractive index of ~1.40. The nominal diameter of the magnetite nanoparticles was 10 nm. Transmission of the TFBG is modulated by the refractive index of the MF, which is sensitive to an external magnetic field.

Taghizadeh et al. [108] proposed a novel magnetic field sensor that is based on the combination of in-line tapered photonic crystal fiber (PCF) Mach–Zehnder interferometer and MNPs, which was theoretically investigated and experimentally realized (Figure 13).

Magnetite-based FF was synthesized using the reverse co-precipitation method and was injected inside the PCF air-holes using a syringe. The effect of the mechanical strain and the magnetic field on the sensitivity of the sensor was studied. The experimental results show that the refractive index changes of the MNP-infiltrated PCF under the applied magnetic field lead to variations in the interferometric output. The results show a very good linear response, which is an essential requirement for the practical sensors. The proposed magnetic field sensor finds applications in various areas, such as optical sensing, military, power industry, and tunable photonic devices.

Li et al. [109] proposed and investigated a novel magnetic field sensor that consists of a U-bent single-mode fiber fixed in a magnetic-fluid-filled vessel. Neither mechanical modification nor additional fiber grating is needed during the sensor fabrication. The results show that the response of the magnetic fluid to the magnetic field can be used to measure the direction and intensity of the magnetic field via whispering gallery modes supported by the U-bent fiber structure with a suitable bending radius. The authors concluded that the proposed sensor has the advantages of easy fabrication and high mechanical strength, which has potential applications in various fields.

The process of magnetic field sensors’ fabrication is shown in Figure 14.

The main ideas of the abovementioned investigations were to combine different shapes of the optical fibers and optical grating with magnetic FFs to fabricate magnetic field sensors. The response of magnetic FFs to the magnetic field was used to measure the intensity of the external magnetic field. Such sensors are easy to fabricate, and they could find application in various fields, such as the power industry, optical sensing, and tunable photonic devices.

### 5.4. “Smart” Windows

Nowadays, “smart” windows have become more and more popular. This technology means that the glass in windows changes its properties from transparent to opaque and vice versa. Some technologies alter the light transmission properties of the windows and are grounded in the use of liquid crystals, polymers, etc. [110] (Figure 15). We believe that the utilization of magnetic FFs for the fabrication of “smart” windows is a promising technology due to the unique properties of FFs. In such technology, the light transmission properties of the windows are altered by a magnetic field. We carefully investigated the current state of the field of magnetic FFs applied in “smart” window fabrication and found only some references that match this criterion.

Heiz et al. [111] presented a switchable, ultrathin suspended particle device (SPD) for large-area integration with smart facades that is based on a fluidic window, rolled glass, and a thin cover with high surface strength. Loading the circulating fluid with nanoscale magnetite nanoparticles of particle sizes ranging from 50 to 100 nm enables active shading and solar-thermal energy harvesting, whereby the loading state and, hence, the optical properties of the liquid can be controlled via remote switching in a particle collector-suspender device. In the fully shaded state, a typical harvesting efficiency of 45% of the incoming solar power is obtained.

Seo et al. [112] reported a facile method for preparing anisotropic magnetic micro-rods dispersed in a liquid mixture that can be utilized for “smart” windows by changing the direction of the magnetic field (Figure 16). The Fe_2_O_3_ nanoparticles used were ~50 nm in diameter. An isopropyl alcohol and polyethylene glycol mixture was used as the dispersion liquid. The rod length was controlled by the nanoparticle concentration, magnetic field application time, and dispersion liquid viscosity. The authors demonstrated that the Fe_2_O_3_ magnetic micro-rods remained stable even after one week. This simple preparation method for the anisotropic microfeatures was concluded to be key in addressing the economical limitations of “smart” window technologies.

We emphasize that current technologies of “smart” windows (based on usage of liquid crystals, polymers, etc.) are only effective in blocking visible light but not infrared radiation, thus warming an interior space. As magnetite nanoparticles absorb and convert heat from infrared irradiation, we conclude that the utilization of magnetite FFs for “smart” window technology could serve as heat capture and provide some advantages in comparison with other “smart” window technologies. The main challenge of magnetite “smart” window technology is the long-term FF stability, as such technology needs FFs to be stable for, at least, dozens of years.

### 5.5. Magnetic Resonance Imaging

Magnetic resonance imaging (MRI) exploits MNPs as an agent used for contrast enhancement. MRI contrast agents could be divided into two categories according to different changes in relaxation times: negative (decreasing the brightness of the image) or T_2_-contrast agents, and positive (increasing the brightness of the image) or T_1_-contrast agents. Magnetic iron oxide NPs are widely used for various biomedical applications, including MRI, due to their biocompatibility and non-toxicity. Magnetite (Fe_3_O_4_) is known as a T_2_-contrast agent, and its application in MRI leads to dark signals, which makes distinguishing naturally occurring hypointense areas in many diseases difficult, so its usage is far from standard in clinical practice [31] (Figure 17).

Thus, intensive research was focused on the development of non-toxic positive contrast agents. Compared with well-studied Gd-based T_1_ and iron oxide nanoparticle-based T_2_ contrast agents, the development of ultrasmall iron oxide-based T_1_ contrast agents is still in its infancy [114]. Further studies of biocompatibility, biodistribution, and pharmacokinetics of ultrasmall iron oxides need to be performed. Nowadays, ferumoxitol is a commercially available and US Food and Drug Administration (FDA)-approved drug for the treatment of iron-deficient anemia, but it could serve as an alternative for gadolinium as a contrast agent [115]. It consists of 3 to 10 nm mixed magnetite/maghemite cores coated with a carbohydrate shell and is shortened to both T_1_ and T_2_. Due to the vast amount of information concerning this topic, we briefly report about recent achievements in this area:

Marashdeh et al. [116] synthesized maghemite (γ-Fe_2_O_3_) NPs of different sizes (22 nm and 30 nm) using the sol–gel method and investigated their potential application as magnetic resonance imaging contrast agents. The relaxation time (T_2_) of the NPs was measured at room temperature. The size was found to affect the contrast enhancement of the MRI image, with the T_2_ for 22 nm-sized γ-Fe_2_O_3_ NPs exhibiting a shorter dephasing compared with the 30 nm-sized γ-Fe_2_O_3_ NPs. T_2_ relaxivity was shown to decrease with an increasing concentration (9–84 μg/mL) of the nanoparticles. Based on the T_2_-weighted analysis, a better signal (i.e., brighter image) was achieved for the 30 nm sized γ-Fe_2_O_3_ nanoparticles. The conclusion was made that synthesized γ-Fe_2_O_3_ nanoparticles are promising materials for use as MRI contrast agents.

C. Tao et al. [117] reported the synthesis of MNPs functionalized with natural bovine serum albumin (BSA) and an artificial poly(acrylic acid)-poly(methacrylic acid (PMAA-PTTM)). They obtained NPs of similar sizes and magnetization using the co-precipitation method and compared their MRI performances. Fe_3_O_4_-BSA was shown to be a darkening contrast enhancement for liver and kidney sites of mice, while Fe_3_O_4_-PMAA-PTTM displayed brighter contrast enhancement for liver and kidney sites. The different MRI behaviors of the synthesized NPs demonstrated that the surface ligands play an important role in optimizing the MRI performance of MNPs. The authors expected that these results would facilitate the design of macromolecule ligands for developing an iron oxide-based T_1_-weight contrast agent. Later on, this group [118] used poly(acrylic acid) (PAA), poly(allylamine hydrochloride) (PAH), and polyvinyl alcohol (PVA), which possess negative, positive, and neutral charges with carboxylic acid, amino, and hydroxyl groups, respectively, as templates and stabilizers to fabricate magnetite nanoparticles using a co-precipitation reaction. The NPs obtained showed slight differences in size and water dispersibility. Moreover, magnetite nanoparticles modified with PAA and PVA showed good biocompatibility, while those modified using PAH displayed high cytotoxicity during cell viability assay. In vitro and in vivo experiments demonstrated that both Fe_3_O_4_-PAA and Fe_3_O_4_-PVA are adequate as T_1_-weighted contrast agents, but Fe_3_O_4_-PAA exhibited a better T_1_ contrast performance (Figure 18).

The macromolecule ligands play an important role in the biocompatibility and T_1_ contrast performance of magnetic Fe_3_O_4_ NPs.

Weia et al. [119] designed and developed zwitterion-coated exceedingly small superparamagnetic iron oxide NPs (ZES-SPIONs) by thermal decomposition of Fe(oleate)_3_ in the presence of oleic acid followed by oxidation with trimethylamine N-oxide. The oxidation step ensures that particles with a maghemite structure are obtained. NPs consist of ~3 nm inorganic cores and ~1 nm ultrathin hydrophilic shells. The ZES-SPIONs synthesized showed high T_1_ contrast power and potential for preclinical MRI and magnetic resonance angiography.

Alipour et al. [120] synthesized a new class of cubic SPIONs as a dual-mode contrast agent in MRI and showed the in vivo feasibility of these NPs as a simultaneous positive and negative contrast agent. SPIONs of two different shapes (cubic and spherical) and three different sizes—7 nm, 11 nm, and 14 nm—were synthesized, and contrast enhancement in vitro was evaluated. Representative transmission electron microscopy images of the NPs synthesized revealed that the particles are well dispersed in a solvent and do not aggregate. By contrast enhancement analysis, among all six SPIONs tested, 11 nm cubic SPIONs were shown to possess optimal contrast enhancement values, which can shorten the spin–lattice and spin–spin relaxation times, simultaneously. It was reported that synthesized NPs are non-toxic. These results demonstrate the promising potential of the synthesized 11 nm silica-coated cubic SPIONs as a synergistic MRI contrast agent.

Li et al. [121] designed and synthesized a novel type of cross-linked iron oxide NP assembly, IONAs, for tumor pH-sensitive T_1_ MR imaging. The cross-linked IONAs were highly stable at a neutral pH even in extremely diluted conditions, which is advantageous over conventional polymer-assisted nanoparticle assemblies that generally lose their integration after dilution. The authors claimed that their study was the first demonstration of an iron oxide nanoparticle-based MR contrast agent that can amplify r_1_ relaxivity in the acidic tumor microenvironment.

Shen et al. [122] synthesized exceedingly small magnetic iron oxide NPs (ES-MIONs) with seven different sizes below 5 nm (i.e., 1.9, 2.6, 3.3, 3.6, 4.2, 4.8, and 4.9 nm) using the co-precipitation method and found that 3.6 nm is the best particle size for ESMIONs to be utilized as a T_1_-weighted MR contrast agent. A drug delivery system based on the 3.6 nm ES-MIONs for T_1_-weighted tumor imaging and chemotherapy was constructed. The authors thus concluded that synthesized nanoparticles are promising for high-resolution T_1_-weighted MR imaging and precise chemotherapy of tumors.

Vangijzegem et al. [123] were focused on the development of very small iron oxide NPs as potential nanoplatforms for the high field (>3 T) T_1_ magnetic resonance angiography (MRA) applications. Magnetite nanoplatforms smaller than 5 nm were prepared by hot injection, and the aqueous transfer was optimized using the introduction of polyethylene glycol (PEG)-based ligands. The as-obtained suspension was shown to be able to be stored for three months in water or saline without any changes in their relaxivities, demonstrating their stability. The concentration-dependent effect was shown using phantom MR images for both T_1_ and T_2_ sequences. The feasibility of the developed system was evaluated in vivo at 9.4 T, demonstrating its usefulness for MRA applications.

Huanga and co-authors [124] reported a new strategy for the fabrication of a single hybrid nanostructure consisting of carbon quantum dots (CQDs), Gd(III) ions, and Fe_3_O_4_ nanoparticles as a new bimodal imaging agent and demonstrated that the nanocomposite could emit fluorescence; thus, it had potential for reliable fluorescence imaging. The r_1_ and r_2_ relaxivities of the nanocomposite were measured to be 5.16 and 115.6 mM^−1^ s^−1^, which nominated the nanocomposite as a suitable T_1_-T_2_ contrast agent. The authors concluded that nanoparticles combining two synergetic imaging modalities showed great potential in FI/MRI dual-modal imaging for a more complementary and accurate detection in medical imaging.

Das and co-authors [125] proposed a new method of fabrication of a single nanoprobe with dual-modality. The nanoprobe was synthesized by co-precipitation of ferrous and ferric salts in the presence of carbon nanodots. The synthesized nanoprobe was shown to be a biocompatible, hemocompatible, and excretable nanostructure, and carbon doping was shown to enhance the r_2_ spin−spin relaxivity of the nanoprobe from 29 to 118.3 mM^−1^ s^−1^, making it a promising T_2_ contrast agent. The authors concluded that carbon dot doping in superparamagnetic iron oxide nanoparticles enhances the area of bioimaging by combining complementary imaging modalities such as fluorescence imaging with MR imaging.

Wang et al. [126] prepared a new type of hybrid nanoparticles integrated with magnetite nanocrystals and carbon dots using a one-pot solvothermal synthesis method. The fabricated nanoparticles demonstrate high magnetic properties (M_s_ = 32.5 A·m^2^/kg) and magnetic resonance imaging ability (r_2_ = 674.4 mM^−1^ s^−1^) from the magnetite nanocrystal core and exhibited intriguing photoluminescent (quantum yield ~6.8%) properties including upconversion fluorescence and excellent photostability from the carbon dots produced in the porous carbon. The authors concluded that mesoporous structures and hydrophilic surface functional groups endow the hybrid nanoparticles with a high drug loading capacity and excellent dispersibility in aqueous solutions.

The investigations of positive (T_1_) non-toxic agents for MRI contrast enhancement as an alternative for the gadolinium contrast agent as well as bimodal (FI/MRI) imaging agents were widely developed. Exceedingly small superparamagnetic iron oxide NPs (up to ~10 nm) with various surface modifications (natural bovine serum albumin, artificial poly(acrylic acid)-poly(methacrylic acid, poly(allylamine hydrochloride)), polyvinyl alcohol, zwitterion-coated, etc.) were shown to have high T_1_ contrast power. Hybrid nanostructures consisting of carbon quantum dots and Fe_3_O_4_ nanoparticles demonstrating biocompatibility showed great potential as bimodal imagine agents.

Despite investigations in the area of the design and synthesis of novel contrast agents for MRI being plentiful, the reproducibility and scalability of the synthesis of uniform NPs with narrow-size distribution as well as functionalization of the surface for a specifical target to different tissues and cells require further investigation.

### 5.6. Solar Energy Harvesting

The ability of MNPs to absorb solar irradiation can also be utilized for the fabrication of new devices for sustainable and renewable energy production. During the last decade, great progress in the creation and fabrication of new photovoltaic devices that could convert solar energy directly to electricity was made. Nanofluids of different origins are utilized for solar thermal applications and heat transfer processes [127], especially magnetic nanofluids.

Sani and co-authors [128] presented a functional fluid consisting of a stable colloidal suspension of maghemite MNPs in water to evaluate its potential for direct electricity generation from the thermoelectric effect enabled by the absorption of sunlight. The ferrofluids studied were composed of maghemite (γ-Fe_2_O_3_) nanoparticles, obtained via the Massart method, and coated with a statistical copolymer made of equimolar acrylic and maleic acid monomers (PAAMA) of average molecular weight 3000 g/mol dispersed in water. The authors investigated the thermoelectric and the optical absorption properties and found that the inclusion of 0.5% of MNPs accelerates the mass transfer rate of the redox couples within the thermocell when heated from the top, which leads to a 2–3 fold increase in the cell’s power output. Moreover, the overall level of optical absorption and its spatial distribution within the nanofluid volume were fully tailorable, acting on the concentration of both MNPs and redox couple ions, allowing for the optimization of future sunlight-enabled thermocell geometry. The authors concluded that these findings demonstrate the nanofluid’s potential as a heat transfer fluid for co-generating heat and power in brand new hybrid flat-plate solar thermal collectors where top-heating geometry is imposed.

Grosu et al. [129] utilized natural magnetite for thermal energy storage. For this purpose, samples of natural magnetite were partially transformed under thermal treatment into hematite, and as a result, the authors obtained a material with excellent thermophysical properties and the ability to control thermal conductivity in a wide range of values. The calculated thermal conductivity for the treated magnetite was very high, and the values obtained were much greater (more than two times) than the average compared with other ceramic materials (concrete, granite, marble, sandstone, etc.) reported that had the potential for thermal energy storage applications. The authors concluded that the combination of such properties is exceptional and crucially advantageous for thermal energy storage applications such as packed-bed heat storage systems.

Kannana et al. [130] fabricated a device for dye-sensitized solar cells based on magnetite nanoparticles. The authors reported the effect of demagnetizing fields caused by magnetite nanoparticles on the efficiency of TiO_2_ based dye-sensitized solar cells. The addition of magnetite nanoparticles to the photoanode enhanced the power conversion efficiency of dye-sensitized solar cells. The authors believed that their work would stimulate future studies on dye-sensitized solar cells devices based on the demagnetization field.

Cao et al. [131] proposed combining three energy storage materials—namely, microporous foam, MNPs, and revolutionary tubes—for the first time to obtain a new flat plate solar collector. The authors claimed that simultaneous use of three different materials in the collector covers each material’s weaknesses. The collector was configured in the following way: a thin metal foam was attached inside the tubes as porous fins to increase the contact area for more heat transfer. The rotation in the tube circulates the nanofluid and pushes it toward the porous fin by a centrifugal force. Such a configuration leads to the recovery of up to 50% of the lost energy and to an increase in energetic performance from ~53% to ~76% in optimal conditions.

Wang and co-authors [132] investigated the influence of magnetite nanoparticles and an external magnetostatic field on the power conversion efficiencies of polymer solar cells. The authors introduced magnetite nanoparticles (with the size of ~5 nm) and orientated them within a bulk heterojunction composite using an external magnetostatic field (Figure 19), which led to over 50% increased efficiency compared with the control polymer solar cell. The authors concluded that the proposed application is an extraordinarily effective way to enhance the power conversion efficiency of polymer solar cells.

Ansari and co-authors [133] reported a low-cost and stable perovskite solar cell using solution-processed nanostructured magnetite as a potential hole transport layer. They confirmed that magnetite nanoparticles could be a potential hole-transporting material candidate for perovskite solar cells. They showed that the fabricated device shows a maximum power conversion efficiency of ~15% and long-term durability up to 30 days.

To conclude, MNPs are promising candidates for applications in solar energy harvesting and in the conversion of solar irradiation into useful energy. However, the large-scale utilization of magnetite nanoparticles for these purposes faces various obstacles, including nanoparticles’ sedimentation and aggregation (dispersion stability), cost of nanomaterials, their toxicity, etc. Therefore, further investigations are needed to find a possible route of application of magnetite nanoparticles in solar energy harvesting and storage.

Summaries of the abovementioned techniques in optics and photonics that utilize magnetite FFs are shown in Table 3.

## 6. Conclusions

Nowadays, ferrofluids are utilized for an extremely wide range of fascinating applications, including new optical devices and imaging techniques. Despite recent progress in the field of synthesis and surface functionalization of iron oxide NPs with tunable sizes and shapes for different applications, numerous challenges still need to be overcome. The most important challenge is obtaining iron oxide NPs of controllable sizes, morphologies, and crystallinities. To obtain photonic materials across the entire visible region, one needs to synthesize NPs with controlled diameters. Most synthesized iron oxide NPs have non-stoichiometric structures, which leads to many complications. A non-stoichiometric structure leads to failure when increasing the absorption band in the near-infrared wavelength region, which restricts its application in the field of optics and photomedicine. Therefore, new strategies for synthesizing MNPs need to be developed to achieve this goal (i.e., synthesis in a constrained environment, bio-inspired synthesis, synthesis using magnetotactic bacteria, etc.). One more challenge is the high oxidation of ultrafine iron oxide NPs due to their large specific surface area and air sensitivity. Long-term stability of iron-oxide-based ferrofluids during storage and application is also critical challenge. Thus, surface coverage and functionalization of iron oxide NPs still need to be optimized for the development of highly stable iron-oxide-based ferrofluids. The synthesis procedures need to be improved for large-scale or industrial production of iron oxide NPs for use in commercial or clinical applications.

Magnetic FFs find their application in optics due to their unique magneto-optical properties such as optical anisotropy, birefringence, and field-dependent transmission. The applications presented in this review include photonic materials, organic light-emitting diodes (OLEDs), magnetic field sensors, “smart” windows, magnetic resonance imaging, and solar energy harvesting. All of these applications utilize the response of FFs to an external magnetic field.

The optical properties of classical FFs can be tuned by applying an external magnetic field with the formation of colloidal photonic crystals that could serve as a new platform for the fabrication of novel optical microelectromechanical systems and sensors. The main challenge of this application is the need for NPs with controlled diameters and stoichiometric structures.

Gold-doped magnetite nanoparticles efficiently enhance luminescence of OLEDs and are very promising for the fabrication of new high-efficient OLEDs with increased light extraction. The following challenges still need to be overcome: controlled size of magnetite nanoparticles and their aggregation.

A combination of different shape optical fibers and optical grating with magnetic FFs allows for the fabrication of magnetic field sensors that use the response of magnetic FFs to a magnetic field for measuring the intensity of the external magnetic field. Such sensors find applications in the power industry, optical sensing, and the design of tunable photonic devices. However, the long-term colloidal stability of FFs needs to be ensured for such applications.

The use of magnetite FFs for “smart” window technology could improve heat capture and shading. The main challenge of magnetite “smart” window technology is also long-term FF stability, as such technology needs FFs to be stable for at least dozens of years.

MNPs are now widely investigated as positive (T_1_) agents for NMI contrast enhancement as an alternative to the gadolinium contrast agent. Exceedingly small superparamagnetic iron oxide NPs (up to ~10 nm) with various surface modifications show high T_1_ contrast power. The main challenges of these applications are its reproducibility, its scalability of the synthesis methods, and its functionalization of the surface for specifically targeting to different tissues and cells.

The ability of capture heat using MNPs has also since been utilized for the creation of new devices for sustainable and renewable energy production based on solar cells. Magnetic FFs substantially increase the power conversion efficiency of existing solar cells. The main challenges of these applications are sedimentation and aggregation of nanoparticles (colloidal stability of the suspension), the cost of nanomaterials, and the toxicity.

Despite the abovementioned challenges, iron-oxide-based magnetic ferrofluids are promising as novel optical and photonic devices and techniques because of their unique properties that can be tuned by an external magnetic field.

## Figures and Tables

**Figure 1 materials-15-02601-f001:**
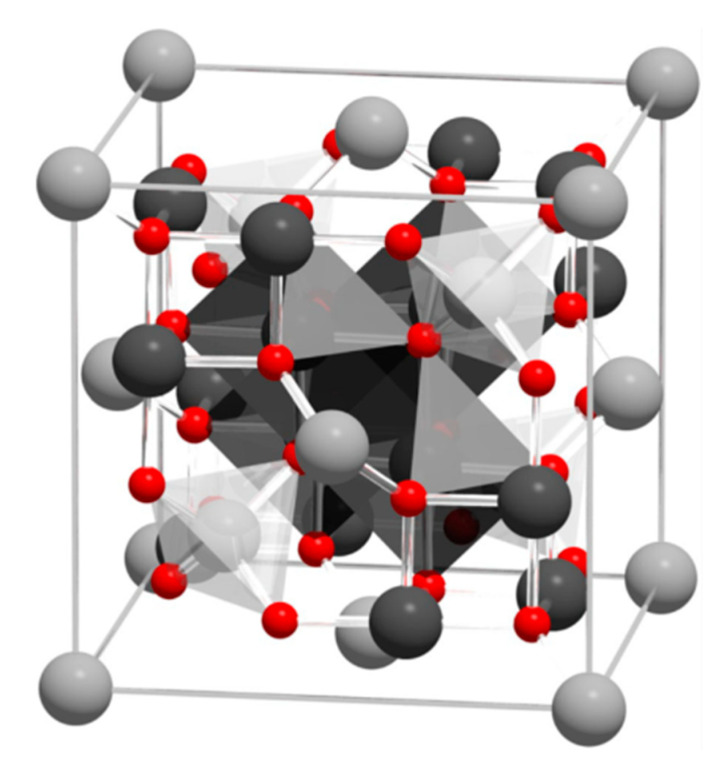
Visualization of the magnetite unit cell identified using octahedral Fe^2.5+^ (dark grey), tetrahedral Fe^2+^ (light grey), and oxygen (red). The local site symmetries are shown by the octahedral and tetrahedral shapes around fully coordinated Fe sites within the unit cell. The different bond angles between the Fe sites lead to dominant antiferromagnetic coupling between the tetrahedral and octahedral sites, giving a bulk ferrimagnetic order (adapted from [34] with permission from Springer Nature).

**Figure 2 materials-15-02601-f002:**
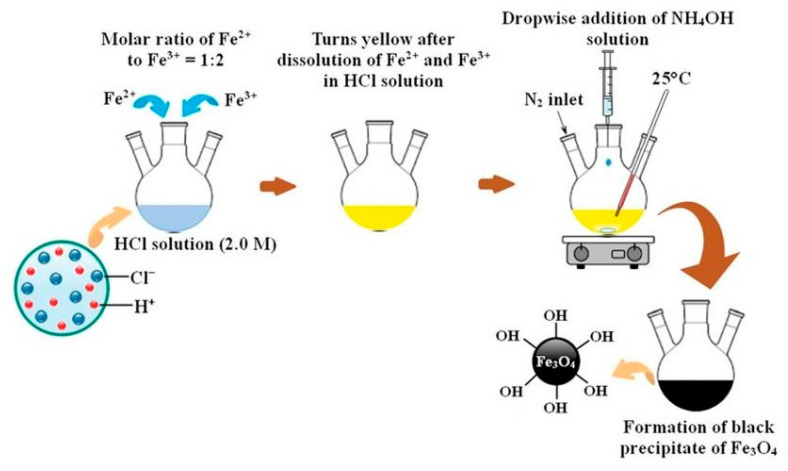
Schematic representation of magnetite synthesis by co-precipitation method (adapted from [44] with permission from the Taylor & Francis Group).

**Figure 3 materials-15-02601-f003:**
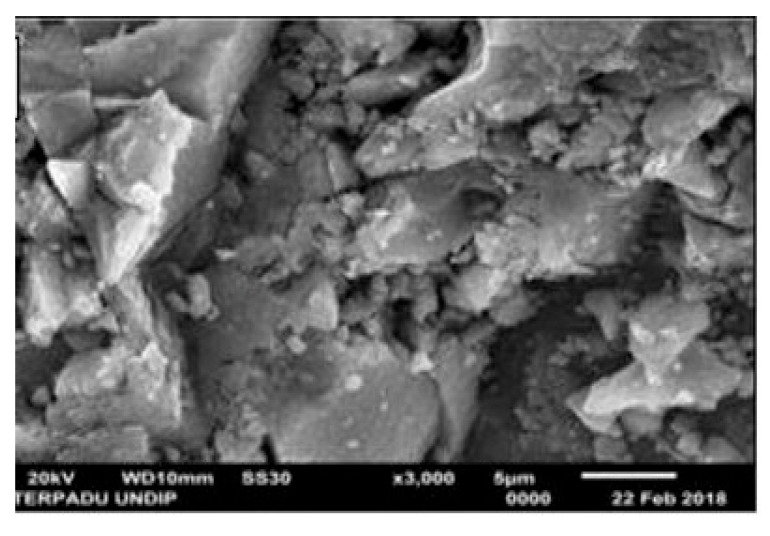
Scanning Electron Microscopy (SEM) image of magnetite nanoparticles synthesized by co-precipitation method (adapted from [41] with permission from IOP Publishing).

**Figure 4 materials-15-02601-f004:**
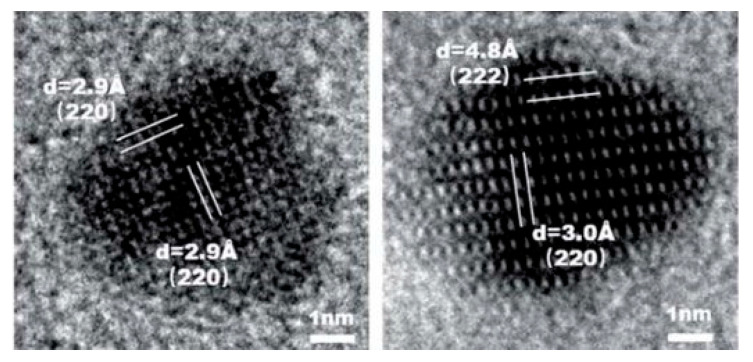
High-resolution Transmission Electron Microscopy (TEM) images showing lattice fringes of the magnetite cores (adapted from [50] with permission from The Royal Society of Chemistry 2019).

**Figure 5 materials-15-02601-f005:**
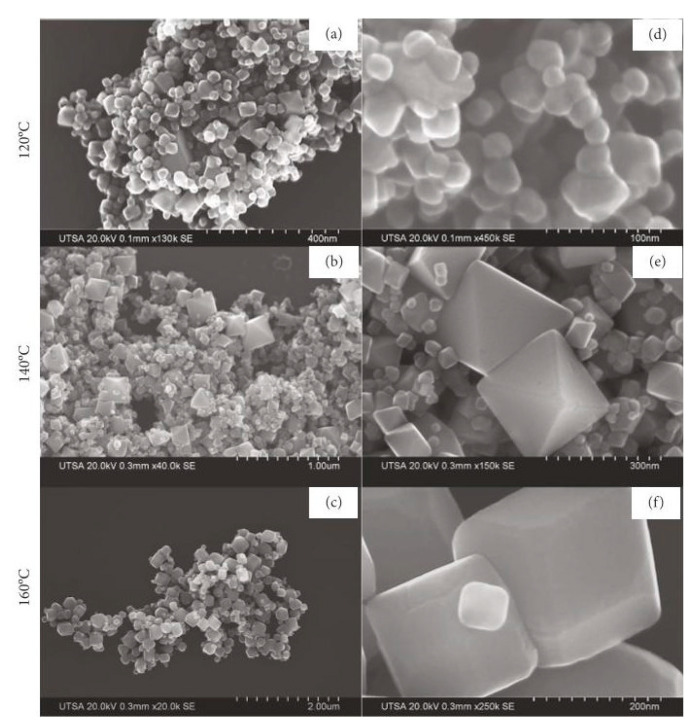
SEM images of magnetite nanoparticles obtained after hydrothermal synthesis at different temperatures: (**a**,**d**) 120 °C, (**b**,**e**) 140 °C, and (**c**,**f**) 160 °C (adapted from [58] with permission from 2019 Nayely Torres-Gómez et al.).

**Figure 6 materials-15-02601-f006:**
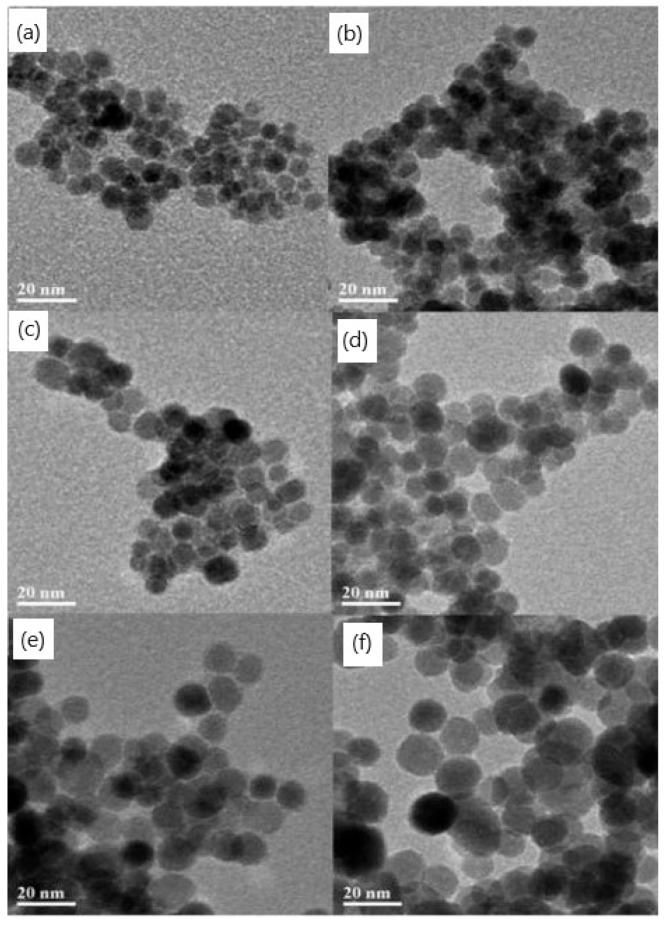
TEM images of iron oxide nanoparticles synthesized using different reaction times in tri(ethylene glycol): (**a**) 1 h, (**b**) 2 h, (**c**) 4 h, (**d**) 8 h, (**e**) 12 h, and (**f**) 24 h (adapted from [63] with permission from the Royal Society of Chemistry).

**Figure 7 materials-15-02601-f007:**
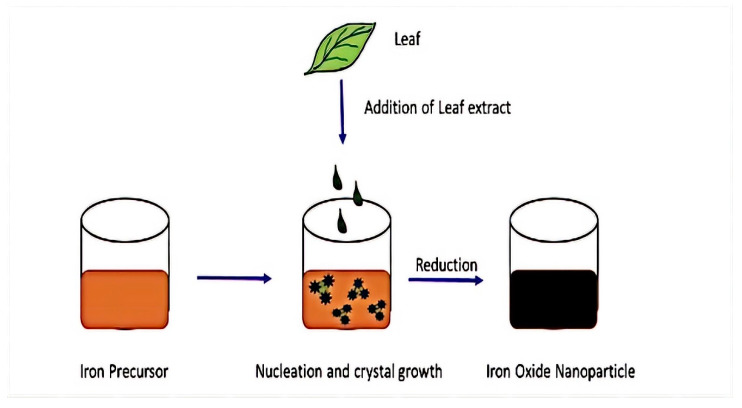
Magnetite nanoparticle synthesis using plant extracts (adapted from [70] with permission from Elsevier).

**Figure 8 materials-15-02601-f008:**
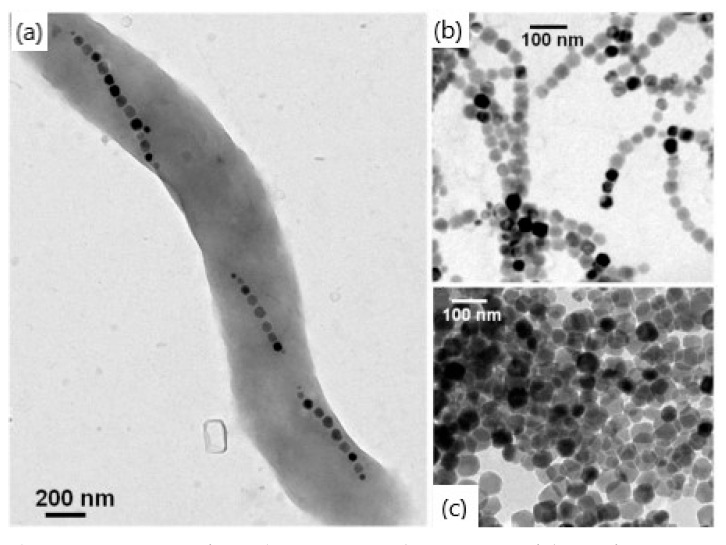
TEM images of a single magnetotactic bacterium (**a**), of chains of magnetosomes extracted from whole magnetotactic bacteria (**b**), and of individual magnetosomes detached from the chains (**c**) (adapted from [74] with permission from 2014 Alphandéry).

**Figure 9 materials-15-02601-f009:**
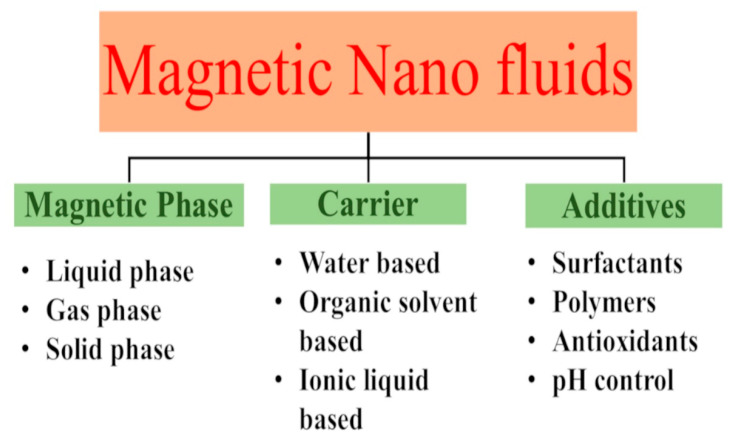
Schematic representation of the preparation of magnetic nanofluids.

**Figure 10 materials-15-02601-f010:**
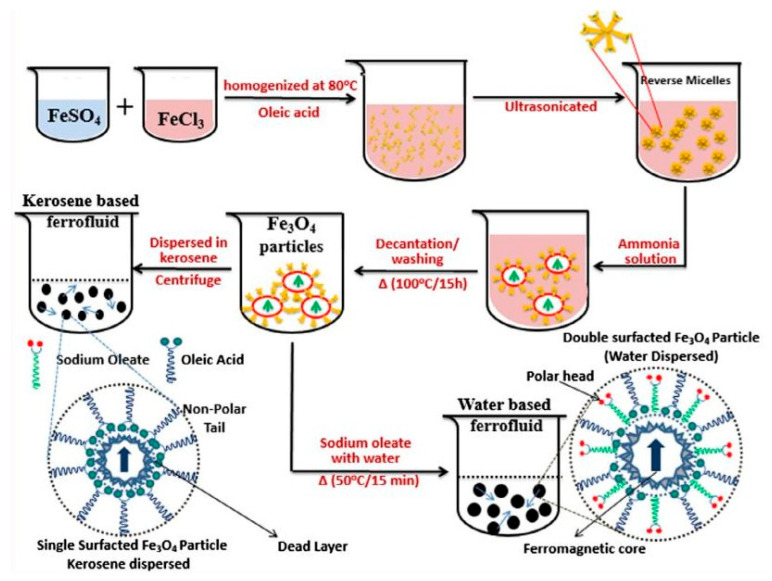
Schematic representation of the aqueous and kerosene-based magnetic fluid preparation by dispersing double surfactant (oleic acid (blue string) and sodium oleate (green string)) Fe_3_O_4_ MNPs using the two-step wet chemical synthesis method (adapted from [90] with permission from Elsevier, 2019).

**Figure 11 materials-15-02601-f011:**
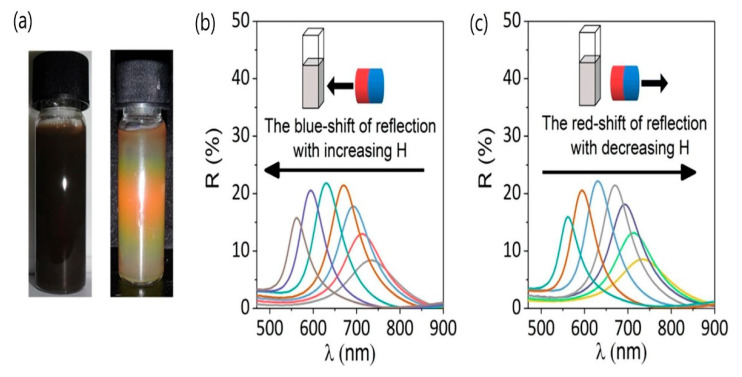
Reversible optical responses of a 100 nm Fe_3_O_4_ colloid under increasing or decreasing external magnetic field (H): (**a**) digital photos of the Fe_3_O_4_ colloid without H (left) and with H (right); (**b**) blue shift in the reflection when H is enhanced; and (**c**) red shift in the reflection when H is weakened (adopted from [100] with permission from Elsevier, 2019).

**Figure 12 materials-15-02601-f012:**
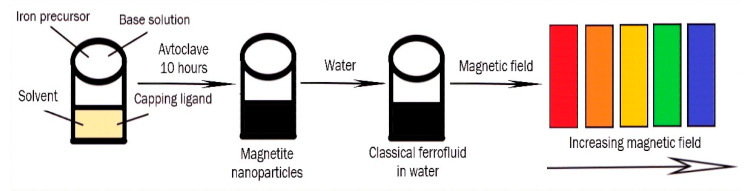
Application of magnetic ferrofluids for the preparation of photonic materials.

**Figure 13 materials-15-02601-f013:**
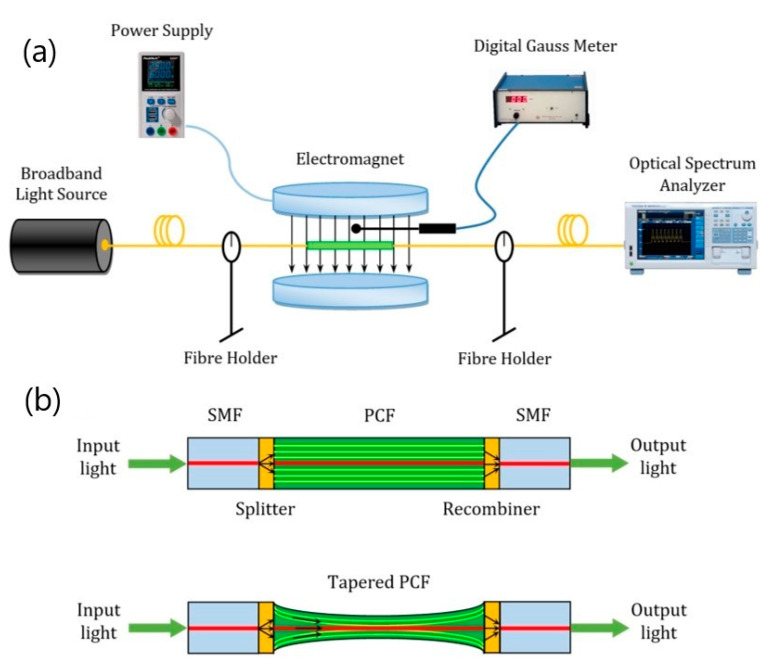
Schematic diagram of (**a**) the experimental setup for magnetic feld sensing and (**b**) the in-line Mach–Zehnder interferometer in tapered photonic crystal fiber (PCF), with single mode fiber (SMF) (adapted from [108] with permission from the Springer Nature).

**Figure 14 materials-15-02601-f014:**
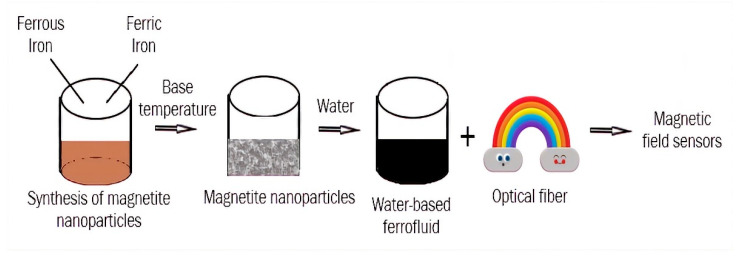
Application of magnetic ferrofluids for fabrication of magnetic field sensors.

**Figure 15 materials-15-02601-f015:**
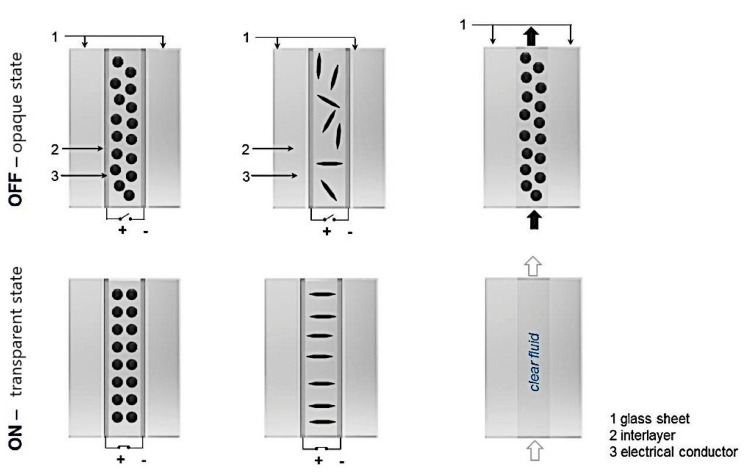
Suspended particle device (SPD) technologies for switchable shading and control of optical transparency. An active fluid is contained within a glass–glass laminate in which an external trigger allows for variable orientation of (**a**) suspended particles or (**b**) liquid crystals. (**c**) A passive fluid with variable transparency flows through a microfluidic device (adapted from [111] with permission from John Wiley and Sons).

**Figure 16 materials-15-02601-f016:**
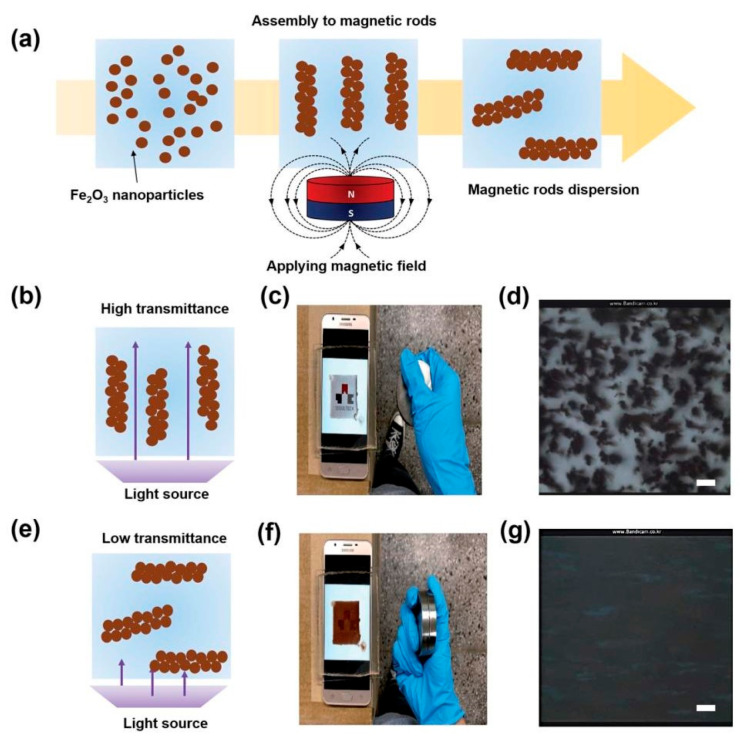
Schematic illustration of the formation of the magnetic microrods by applying a magnetic field (**a**). Schematic illustration of magnetic rods vertically oriented by controlling the magnetic field (**b**). Smart phone screen showing the university logo through the transparent cavity filled with magnetic rods (**c**). Microscopic image of the vertically oriented magnetic rods (**d**). Schematic illustration of magnetic rods oriented parallel to the surface by controlling the magnetic field (**e**). Smart phone screen showing the university logo through the transparent cavity filled with magnetic rods when the rods are in parallel to the surface (**f**). Microscopic image of the parallel oriented magnetic rods (**g**). The scale bars represent 100 μm (adapted from [112] with permission from the Royal Society of Chemistry).

**Figure 17 materials-15-02601-f017:**
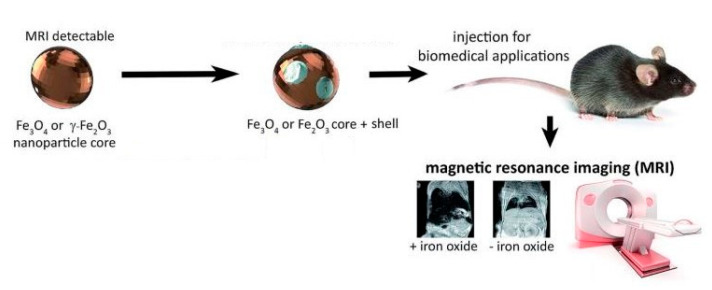
Concept of iron oxide NP application in MR imaging (adopted from [113] with permission from the Royal Society of Chemistry).

**Figure 18 materials-15-02601-f018:**
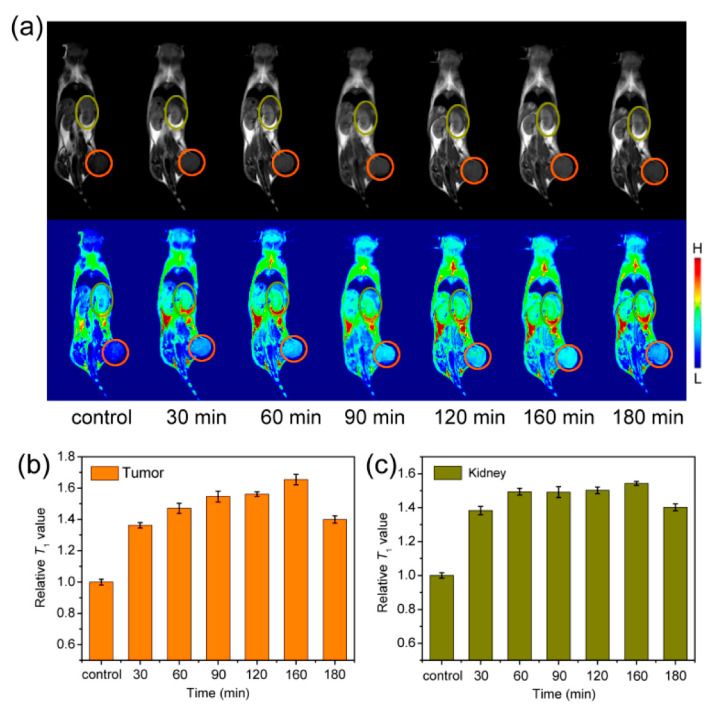
T_1_-weighted MR images (B_0_ = 1 T) of mice collected before (control group) and after intravenous injection of Fe_3_O_4_-PAA at time points of 30, 60, 90, 120, 160, and 180 min (**a**). The corresponding relative T_1_-weighted signals extracted from (**b**) tumor (orange circle) and (**c**) kidney (dark yellow circle) sites (adapted from [118]).

**Figure 19 materials-15-02601-f019:**
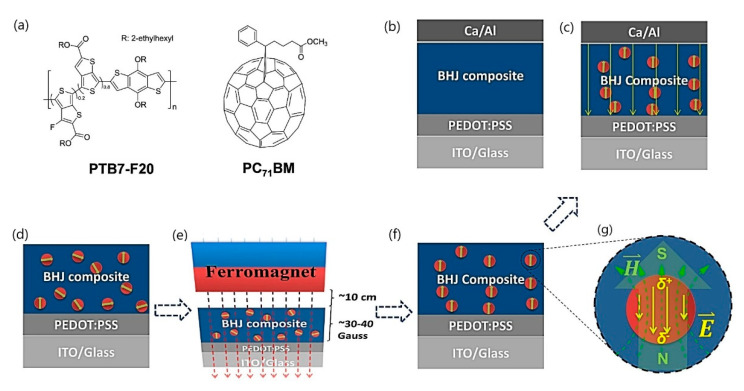
The molecular structures of PTB7-F20 and PC71BM (**a**); the conventional device structure of PSCs without incorporating any Fe_3_O_4_ magnetic nanoparticles (MNPs) (**b**); the conventional device structure of PSCs incorporated with Fe_3_O_4_ MNPs and aligned by an external magnetostatic field (**c**); the fabrication procedures of PSCs incorporated with Fe_3_O_4_ MNPs and aligned by an external magnetostatic field (**d**)–(**f**); BHJ active layer incorporated with Fe_3_O_4_ MNPs was spin-coated on a PEDOT:PSS-coated ITO substrate (**d**); and a ferromagnet was suspend above the surface of BHJ composite incorporated with the Fe_3_O_4_ MNP layer. The magnetic intensity was ~30–40 G, and the distance between the ferromagnet and BHJ composite layer was ~10 cm (**e**); oriented Fe_3_O_4_ MNPs inside BHJ active layer by an external magnetostatic field. In pre-devices (**f**), a drawing of a partial enlargement of the Fe_3_O_4_ MNP in (**c**), showing an antiparallel relation between the magnetic dipole (caused by the Fe_3_O_4_ crystal inside the particle) and electric dipole (caused by the difference in charge density between the inside Fe_3_O_4_ and outside organic coater) (**g**) (adapted from [132] with permission from Scientific Reports).

**Table 1 materials-15-02601-t001:** Properties of magnetite.

Properties	Magnetite
Molecular formula	Fe_3_O_4_
Crystal structure	Cubic
Density (g/cm^3^ )	5.18
Melting point (°C)	1583–1597
Boiling point (°C)	2623
Color	black
Hardness	5.5
Type of magnetism	ferrimagnetic
Curie temperature (K)	580
Ms at 300 K (A·m^2^/kg)	92–100
Magnetism (nanoparticles)	Superparamagnetic
Lustre	Metallic
Diaphaneity	Opaque
Crystal System	Isometric
Birefringence	Isotropic minerals have no birefringence
Refractive Index values	*n* = 2.42

**Table 2 materials-15-02601-t002:** Magnetic nanoparticles (MNP) synthesis and their comparison in terms of synthetic routes, advantages, disadvantages, and challenges.

Method		Conditions	Advantages	Disadvantages	Challenges
Chemical routes	Co-precipitation	Co-precipitation of ferrous and ferric salts in water with a strong basic solution at room temperature or higher	Simple, large quantity of nanoparticles	Poor morphology and non-stoichiometric magnetite	Size- and shape-controlled synthesis with reproducibility
	Partial oxidation of ferrous hydroxide	Precipitation of amorphous ferrous hydroxide from a ferrous sulphate solution with subsequent aging at 90 °C in the presence of nitrate ion	Versatility and hydrophilic particles	Goethite formation and extensive nanoparticle agglomeration	High oxidation of ultrafine iron oxide nanoparticles and aggregation of nanoparticles
	Reaction in constrained environments	Precipitation of iron oxides using synthetic and biological reactors (apoferritin protein cages, micelles, mesoporous templates, and microemulsion)	Nanoparticles with uniform dimensions preventing oxidation and particle interaction	Complicated conditions	Purification of nanoparticles and scale-up procedures
	Hydrothermal or high-temperature reactions	Wet-chemical method of magnetite nanoparticles obtained using reactor or autoclave, with a pressure of >6000 Pa and a temperature of >200 °C	Nanoparticles of different shapes and morphologies with high crystallinity	High temperatures and pressures	Need expensive facilities
	Polyol method	High-temperature decomposition of ferric precursors in the polyol medium	Low cost, hydrolytic stability, and nanoparticles with controlled shape and size	Usage of toxic solvents	Thermal instability and flammability
	Sol–gel synthesis	Wet-chemical process based on hydrolysis and polycondensation of iron precursors with the formation of a “sol” and further drying (“gel” formation)	Monodispersity, good control of particle size, control of microstructure, desirable shape and length of the products, high purity, and good crystallinity	Long completion time and toxic organic solvents	Contamination of the product with the matrix component
	Sonochemical synthesis	Chemical reaction occurs due to the application of ultrasound irradiation, which causes acoustic cavitation in the aqueous solutions	Nanoparticles with high crystallinity, saturation magnetization, and narrow size distribution	Shape and size of nanoparticles obtained are difficult to control	Mechanism of reaction is still not well understood
Biological routes	Bacterial	Precipitation of magnetite nanoparticles inside a bacterial magnetosome	Non-toxic, biocompatible, and unique crystal shape	Culturing microorganisms takes more time and complicated equipment	Limited knowledge about biomineralization process and time-consuming
	Plants	Mixing precursor salts with green substrates that act as reducing and limiting agents	Non-toxic, biocompatible, and eco-friendly	Difficulty controlling size and properties	Limited knowledge about the mechanism of the process and time-consuming

**Table 3 materials-15-02601-t003:** Magnetic ferrofluids (FF) applications, descriptions, and challenges in optics and nanophotonics.

Application	Description	Challenges
Photonic materials	Superparamagnetic magnetite nanocrystals or nanoclusters with tunable sizes from 30 to 200 nm in water	NPs with controlled diameters and stoichiometric structures
Organic light-emitting diodes (OLEDs)	Gold-doped magnetite nanoparticles	Size of magnetite nanoparticles and aggregation of nanoparticles
Magnetic field sensors	Combination of different shape of optical fibers and optical grating with magnetic FFs	Long-term colloidal stability of FFs
“Smart” windows	Nanoscale magnetite nanoparticles from 50 to 100 nm used for shading and the optical properties of the window controlled through remote switching of magnetic field	Long-term colloidal stability of FFs
Magnetic resonance imaging	Exceedingly small superparamagnetic iron oxide NPs (up to ~10 nm) with various surface modifications	Reproducibility, scalability of synthesis, and functionalization of the surface for specifically targeting to different tissues and cells
Solar energy harvesting	Sensitized solar cells based on MNPs	Nanoparticle sedimentation and aggregation, cost of nanomaterials, and toxicity

## Data Availability

The data presented in this review are available from the original publications that are cited.

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
