# Peer review of "Magnetite Nanoparticles: Synthesis and Applications in Optics and Nanophotonics"

_materials, 2022, doi:10.3390/ma15072601_

Round 1
Reviewer 1 Report
This work presents a timely review of the synthesis of magnetite nanoparticles and magnetic nanofluids and their various applications. It is informative and well organized. The major drawback of the current manuscript is the English language. I listed some of them but the list is far from complete. I recommend its publication in Materials once the English language is polished and my other comments given below are addressed.
1) The authors described synthesis methods using a large portion of the text. However, the authors didn’t mention synthesis in the title at all. The authors may consider modifying the title to better reflect the content of the paper.
2) In the last paragraph of Section 2, the authors owed the magnetic ordering of magnetite to the exchange interaction, which is invalid. Magnetocrystalline anisotropy plays a critical role in the magnetic ordering of magnetite. An account of magnetocrystalline anisotropy needs to be added.
3) References need to be given for different reactors listed in lines 135—136.
4) The following typos need to be corrected.
4.1) In line 59, the comma in “0,5” should be a decimal point “.”.
4.2) In lines 61—64, 96, Fe^{+2} needs to be changed to Fe^{2+} at multiple places. Similarly for Fe^{+3}, Mg^{+2}, Co^{+2}, Ni^{+2}, and O^{-2}.
4.3) In line 77, the multiplication sign * in “A*m2/kg” is mainly used in computer languages. It is better to use the dot operator “⋅” or the sign “×”.
5) The following is an incomplete list of English language errors.
5.1) A verb is missing for the sentence “Among them NPs ..” in lines 31—32.
5.2) A verb is missing for the sentence “MNPs thought to be the most …” in lines 36—38.
5.3) A verb is missing for the sentence “Magnetite, Fe3O4, …” in line 55.
5.4) A verb is missing for the sentence “The unique magnetic …” in lines 69—71.
5.5) In line 81, the expression “high magnetic properties” is ambiguous. Do you mean high magnetic moment, robust magnetic properties, or something else?
5.6) In line 99, the preposition “at” should be “in”.
5.7) The comma at the end of Eqs. (2)-(5) should be changed to the period.
5.8) In line 124, the final size is one of the properties of MNPs. The authors can either delete “and final size” or present the final size as an example.
5.9) In line 124, the two commas are unnecessary.
5.10) In line 125, it is better to extend “pH” to “the pH value of X” where X needs to be specified by the authors for clarity.
5.11) The first word “The” in line 134 needs to be deleted.
5.12) An article is needed for the sentences “Ferritin is iron-storage protein” and “it has spherical protein cage” in line 140.
5.13) The authors may consider spelling out the author name for “Authors [44]” in line 141.
5.14) A white space is missing between “et” and “al.” in line 145, and “et al.” should be italic.
5.15) The word “prevent” in lines 151—152 should be changed to “preventing”.
5.16) An article is missing for the sentence “It is unique biomineralization system …” in line 152.
5.17) In line 517, the word “been” needs to be deleted.
5.18) In line 571, the word “it” should be changed to “its”.
Author Response
First, we would like to thank the reviewer for his positive opinion and for supporting the publication of this paper. Second, we improved the English language of the Manuscript.
- “The authors described synthesis methods using a large portion of the text. However, the authors didn’t mention synthesis in the title at all. The authors may consider modifying the title to better reflect the content of the paper.”
Answer: We thank the reviewer for his comment and we modified the title. The new title is “Magnetite nanoparticles: synthesis and applications in optics and nanophotonics”.
- “In the last paragraph of Section 2, the authors owed the magnetic ordering of magnetite to the exchange interaction, which is invalid. Magnetocrystalline anisotropy plays a critical role in the magnetic ordering of magnetite. An account of magnetocrystalline anisotropy needs to be added.“
Answer: We thank the reviewer for his comment and following the reviewer's comment we have added account of magnetocrystalline anisotropy.
- “References need to be given for different reactors listed in lines 135—136.“
Answer: We thank the reviewer for his comment, and following the reviewer's comment we added references needed.
- “The following typos need to be corrected.”
Answer: We thank the reviewer for his comment and corrected all typos listed.
- “The following is an incomplete list of English language errors.”
Answer: We thank the reviewer for his comment and corrected language errors.

Reviewer 2 Report
The paper under consideration is weak and does not qualify to be considered for publication at this stage. However, I have the following suggestions to improve the quality of the manuscript.
- Line 47-53, mention the section numbers to better gives an idea that which topic is discussed in which section. And instead of using future tense, present perfect tense should be used.
- Introduction section is quite small for a review paper. More literature review and history of the topic should be discussed.
- Line 55, “Magnetite, Fe3O4, iron oxide with inverse spinel structure. ” What does this sentence mean?
- In section 2, the structure of Fe3O4 should be drawn. Moreover, make a table for the properties of magnetite.
- The paper is full of English grammar mistakes. Moreover, the sentence structures are also not acceptable. I suggest the author using the English editing service. These are the few mistakes which I am highlighting “The most commonly used method is co-precipitation method firstly reported be Massart…”---- Instead of “be” should be “by”. “Many authors [28, 29, 30, 31, 32] have reported the features of this chemical techniques…” instead of “this” should be “these”. “Magnetite nanoparticles are obtaining by co-precipitation of ferrous and..” instead of “obtaining” should be “obtained”.
- All the figures have poor quality. Not acceptable. Author need to provide high-quality images.
- The article lack figures related to the applications explained in the manuscript. This makes the manuscript visually unattractive and difficult to understand the main idea of the discussion. I suggest the author add more figures in each sections especially in the section 5.
- The author has simply added the information from previous works that XXX et al. presented XXX method. However, there is no critical or conclusive discussion on the literature. Which application is better and their efficiencies? There are no table for comparison, etc.
- How this article stands out from previous review articles on this topic? An explanation is needed to defend the novelty of this review.
- The paper is not well-balanced. Almost half of the paper is written on the synthesis method and then other half on its applications in optics. Therefore, the title of the paper should be modified accordingly.
- Author should make a comparison table to define which synthesis method is cost-effective, simple, and efficient.
Author Response
First, we would like to thank the reviewer for his important comments and questions that enabled us to provide a better and clearer version of the paper.
- “Line 47-53, mention the section numbers to better gives an idea that which topic is discussed in which section. And instead of using future tense, present perfect tense should be used.”
Answer: We thank the reviewer for his comments. We mentioned the section numbers and used present tense.
- “Introduction section is quite small for a review paper. More literature review and history of the topic should be discussed.”
Answer: We thank the reviewer for his comment. We have added more literature and history of the topic.
- “Line 55, “Magnetite, Fe3O4, iron oxide with inverse spinel structure. ” What does this sentence mean?”
Answer: We thank the reviewer for his question and apologize for the lack of clarity. This means that magnetite has the inverse spinel crystal structure. We have fixed this.
- “In section 2, the structure of Fe3O4 should be drawn. Moreover, make a table for the properties of magnetite.”
Answer: We thank the reviewer for his very good comment. We have added the figure that describes the magnetite structure and also the Table that describes magnetite’s properties.
- “The paper is full of English grammar mistakes. Moreover, the sentence structures are also not acceptable. I suggest the author using the English editing service. These are the few mistakes which I am highlighting “The most commonly used method is co-precipitation method firstly reported be Massart…”---- Instead of “be” should be “by”. “Many authors [28, 29, 30, 31, 32] have reported the features of this chemical techniques…” instead of “this” should be “these”. “Magnetite nanoparticles are obtaining by co-precipitation of ferrous and..” instead of “obtaining” should be “obtained”.”
Answer: We thank the reviewer for his comment. We improved the English language of the Manuscript.
- “All the figures have poor quality. Not acceptable. Author need to provide high-quality images.”
Answer: We thank the reviewer for his comment and the quality of figures was improved.
- “The article lack figures related to the applications explained in the manuscript. This makes the manuscript visually unattractive and difficult to understand the main idea of the discussion. I suggest the author add more figures in each sections especially in the section 5.”
Answer: We thank the reviewer for his comment and added more figures in each section.
- “The author has simply added the information from previous works that XXX et al. presented XXX method. However, there is no critical or conclusive discussion on the literature. Which application is better and their efficiencies? There are no table for comparison, etc.”
Answer: We thank the reviewer for his comment and apologize for the lack of clarity. Following the reviewer's comment, we added the discussion on this topic and added the table for comparison of mentioned applications.
- “How this article stands out from previous review articles on this topic? An explanation is needed to defend the novelty of this review.”
Answer: We thank the reviewer for his comment and the explanation about the novelty of the review was added.
- “The paper is not well-balanced. Almost half of the paper is written on the synthesis method and then other half on its applications in optics. Therefore, the title of the paper should be modified accordingly.”
Answer: We thank the reviewer for his comment and the title of the paper was modified accordingly.
- “Author should make a comparison table to define which synthesis method is cost-effective, simple, and efficient.”
Answer: We thank the reviewer for his comment and apologize for the lack of clarity. Following the reviewer's comment, we added the table for comparison of synthesis methods and some recommendation concerning the choice of the synthesis method.

Reviewer 3 Report
In this mini-review titled "Magnetite nanoparticles in optics and nanophotonics" by Dudchenko and coworkers, the authors provided an interesting and timely overview of nanomagnetic particle synthesis and properties. They structured this not so comprehensive review in a way that firstly focused on discussions and comparisons among a variety of synthesis and characterization methodologies of magnetic nanoparticles proposed and developed in the field, and then overviewed various applications in optics and photonics based upon their properties. Their discussions not only summarized the major achievements, progress and challenges on the subject, but also pointed out advantages and disadvantages of each subtopic covered in this mini-review. All of these make this survey a great resource for people who are interested in the topic to know the current status of the subject. From this point of view, this review can serve as a great reference in the field.
After carefully going through the manuscript, I would like to recommend it for publication in Materials as it presents a timely overview of the subject in the field. However, before its formal acceptance, I have few minor suggestions for the authors to address in the revision.
- English. I would strongly suggest the authors to improve the quality of the manuscript by polishing English. As far as I can see, there are many grammar issues throughout the content. Among these issues, most appear as the singular and plural problems, referential problems, etc. As an example of the singular and plural issues, in line 10 "is" should be "are". For the referential issues, one example is "which" in line 25 to be changed as "whose". Besides, the manuscript can become more readable by adding "a", "an", or "the" to many appropriate sentences.
- I noted that some of decimal points "." were mistyped as comma ",". For examples, the decimal points "," appearing in lines 58, 59, and 702 should be changed to ".".
- Another minor suggestion is whether it is possible to make a table by comparing different synthesis methods discussed in Session 3 in terms of their own conditions, advantages, disadvantages, and challenges. A table of such is expected to provide a more straightforward and intuitive way for a reader to comprehend and digest each synthetic method concentrated on in this mini review.
Author Response
First, we would like to thank the reviewer for his positive opinion and for supporting the publication of this paper.
- “English. I would strongly suggest the authors to improve the quality of the manuscript by polishing English. As far as I can see, there are many grammar issues throughout the content. Among these issues, most appear as the singular and plural problems, referential problems, etc. As an example of the singular and plural issues, in line 10 "is" should be "are". For the referential issues, one example is "which" in line 25 to be changed as "whose". Besides, the manuscript can become more readable by adding "a", "an", or "the" to many appropriate sentences.”
Answer: We thank the reviewer for his comment. We improved English taking in account the reviewer’s recommendations.
- “I noted that some of decimal points "." were mistyped as comma ",". For examples, the decimal points "," appearing in lines 58, 59, and 702 should be changed to ".".”
Answer: We thank the reviewer for his comment. We corrected all typos listed.
- “Another minor suggestion is whether it is possible to make a table by comparing different synthesis methods discussed in Session 3 in terms of their own conditions, advantages, disadvantages, and challenges. A table of such is expected to provide a more straightforward and intuitive way for a reader to comprehend and digest each synthetic method concentrated on in this mini review.”
Answer: We thank the reviewer for his comment. The table with comparison of different synthesis methods in terms of their conditions, advantages, disadvantages, and challenges was added.

Reviewer 4 Report
REFEREE REPORT
on paper “Magnetite nanoparticles in optics and nanophotonics”
by authors Nataliia Dudchenko, Shweta Pawar, Ilana Perelshtein and Dror Fixler,
submitted to Materials
The paper “Magnetite nanoparticles in optics and nanophotonics” is devoted to overview of magnetite nanoparticles synthesis and properties. The authors analyzed a big number of relevant publications. The topic of this paper is critically actual especially in medicine, optics, biology etc. The data are reliable and do not cause much doubt. Nevertheless, there are several points before the paper can be published. I hope that authors after major revisions can improve the paper and can publish it in Materials.
- The Introduction part must be improved with literature in the field of magnetic nanoobjects (particles, pillars, tubes, wires etc.) and I suggest to use the following reference (see and discuss:
doi:10.4028/www.scientific.net/SSP.299.100;
doi:10.1038/s41893-019-0452-6;
doi: 10.1039/d0ra07529a).
- The information about crystal structure of the magnetite should be added to the 2 part “Structure and properties of magnetite”. I believe that the Figure with the crystal structure will improve the present review.
- The second part of the manuscript titled “Structure and properties of magnetite” and sounds wide, but this part reports only about crystal structure, magnetic and electron properties. It should be widened.
- Pages 2-6 reports about technological approaches for the magnetite nanoparticles formation but the practical recommendations about the most suitable method for this purpose didn’t suggest.
- The number of figures is too small (7!) for the review. I think that Figures with the morphology investigation of the nanoparticles will improve it. Obviously, the SEM or TEM pictures of magnetite nanoparticles obtained using various technological methods will decorate the review.
- The Conclusion part is too short, please improve it.
- The list of References is formatted not in the MDPI Materials requirements. Please use the necessary citation style via Mendeley (the names of journals should be acronyms).
- There are some insufficient typos and English mistakes in the text.
Author Response
First, we would like to thank the reviewer for his positive opinion and for supporting the publication of this paper.
- “The Introduction part must be improved with literature in the field of magnetic nanoobjects (particles, pillars, tubes, wires etc.) and I suggest to use the following reference (see and discuss: doi:10.4028/www.scientific.net/SSP.299.100; doi:10.1038/s41893-019-0452-6; doi: 10.1039/d0ra07529a).”
Answer: We thank the reviewer for his comment. We improved Introduction part with literature in account the reviewer’s recommendations.
- “The information about crystal structure of the magnetite should be added to the 2 part “Structure and properties of magnetite”. I believe that the Figure with the crystal structure will improve the present review.”
Answer: We thank the reviewer for his comment. We added the Figure with crystal structure of magnetite and some comments on it.
- “The second part of the manuscript titled “Structure and properties of magnetite” and sounds wide, but this part reports only about crystal structure, magnetic and electron properties. It should be widened.”
Answer: We thank the reviewer for his comment. The table which describes the properties of magnetite was added.
- “Pages 2-6 reports about technological approaches for the magnetite nanoparticles formation but the practical recommendations about the most suitable method for this purpose didn’t suggest.”
Answer: We thank the reviewer for his comment. We added information (Table) of advantages and disadvantages of each method listed and some recommendation concerning the choice of the synthesis method.
- “The number of figures is too small (7!) for the review. I think that Figures with the morphology investigation of the nanoparticles will improve it. Obviously, the SEM or TEM pictures of magnetite nanoparticles obtained using various technological methods will decorate the review.”
Answer: We thank the reviewer for his comment. We added the Figures of magnetite nanoparticles obtained and some illustration of their applications in optics and nanophotonics.
- “The Conclusion part is too short, please improve it.”
Answer: We thank the reviewer for his comment. The Conclusion part was improved.
- “The list of References is formatted not in the MDPI Materials requirements. Please use the necessary citation style via Mendeley (the names of journals should be acronyms).”
Answer: We thank the reviewer for his comment. The list of References was formatted according to MDPI Materials requirements.
- “There are some insufficient typos and English mistakes in the text.”
Answer: We thank the reviewer for his comment. We corrected typos and English mistakes.

Round 2
Reviewer 2 Report
I am not satisfied with the revisions offered by the author. I suggest the author providing the data/information they have added in the paper should be presented in the reviewer's report. I am unable to find the queries asked by the reviewer have been answered in the revised version.
For instance:
- “The paper is not well-balanced. Almost half of the paper is written on the synthesis method and then other half on its applications in optics. Therefore, the title of the paper should be modified accordingly.”
Answer: We thank the reviewer for his comment and the title of the paper was modified accordingly.
The author has not modified the title of the paper.
- “How this article stands out from previous review articles on this topic? An explanation is needed to defend the novelty of this review.”
Answer: We thank the reviewer for his comment and the explanation about the novelty of the review was added.
Where is the explanation for this point?
- “The paper is full of English grammar mistakes. Moreover, the sentence structures are also not acceptable. I suggest the author using the English editing service. These are the few mistakes which I am highlighting “The most commonly used method is co-precipitation method firstly reported be Massart…”---- Instead of “be” should be “by”. “Many authors [28, 29, 30, 31, 32] have reported the features of this chemical techniques…” instead of “this” should be “these”. “Magnetite nanoparticles are obtaining by co-precipitation of ferrous and..” instead of “obtaining” should be “obtained”.”
Answer: We thank the reviewer for his comment. We improved the English language of the Manuscript.
Where is the English editing certificate?
Therefore, I am giving one more chance to the authors to provide all the answers properly in the reviewer's sheet so that its visible to the reviewer and assists the positive marking of the paper.
Author Response
We are thankful to the reviewer for sharing his valuable opinion on our work. The comments truly helped us to improve the content. We tried our level best to answer the comments of the reviewer through this revised form of the manuscript.
Answer to Reviewer 2
First, we would like to thank the reviewer for his important comments and questions that enabled us to provide a better and clearer version of the paper.
- “Line 47-53, mention the section numbers to better gives an idea that which topic is discussed in which section. And instead of using future tense, present perfect tense should be used.”
Answer: We thank the reviewer for his comments. We mentioned the section numbers and used present tense (Page 2, Lines 69-75).
- “Introduction section is quite small for a review paper. More literature review and history of the topic should be discussed.”
Answer: We thank the reviewer for his comment. We have added more literature and history of the topic (Pages 1-2).
- “Line 55, “Magnetite, Fe3O4, iron oxide with inverse spinel structure. ” What does this sentence mean?”
Answer: We thank the reviewer for his question and apologize for the lack of clarity. This means that magnetite has the inverse spinel crystal structure. We have fixed this (Page 2, Section2).
- “In section 2, the structure of Fe3O4 should be drawn. Moreover, make a table for the properties of magnetite.”
Answer: We thank the reviewer for his very good comment. We have added the figure that describes the magnetite structure and also the Table that describes magnetite’s properties (Page 3, Section 2).
- “The paper is full of English grammar mistakes. Moreover, the sentence structures are also not acceptable. I suggest the author using the English editing service. These are the few mistakes which I am highlighting “The most commonly used method is co-precipitation method firstly reported be Massart…”---- Instead of “be” should be “by”. “Many authors [28, 29, 30, 31, 32] have reported the features of this chemical techniques…” instead of “this” should be “these”. “Magnetite nanoparticles are obtaining by co-precipitation of ferrous and..” instead of “obtaining” should be “obtained”.”
Answer: We thank the reviewer for his comment. The English language of the Manuscript was checked by an English-editing service. The certificate is attached.
- “All the figures have poor quality. Not acceptable. Author need to provide high-quality images.”
Answer: We thank the reviewer for his comment and the quality of figures was improved.
- “The article lack figures related to the applications explained in the manuscript. This makes the manuscript visually unattractive and difficult to understand the main idea of the discussion. I suggest the author add more figures in each sections especially in the section 5.”
Answer: We thank the reviewer for his comment and added more figures in each section.
- “The author has simply added the information from previous works that XXX et al. presented XXX method. However, there is no critical or conclusive discussion on the literature. Which application is better and their efficiencies? There are no table for comparison, etc.”
Answer: We thank the reviewer for his comment and apologize for the lack of clarity. Following the reviewer's comment, we added the discussion on this topic. On the topic of photonic materials (Page 17, Lines 531-538): “To conclude, a wide range of magnetite nanoparticles were used for the fabrication of photonic materials. Superparamagnetic magnetite nanocrystals or nanoclusters with tunable sizes from 30 to 200 nm, long-term stability and rather high saturation magnetization could be directly utilized for constructing of colloidal photonic crystals through the application of a magnetic field. The optical properties of such nanoclusters can be tuned by applying an external magnetic field (typically 50–500 Oe), and the reflexion spectra of synthesized superparamagnetic colloids have a wavelength range of approximately 420–800 nm, which includes the entire visible range.”. On the topic of OLEDs (Page 18, Lines 566-571): “Organic light-emitting diodes (OLEDs) are usually fabricated using organic molecules or conjugated polymers. During recent years, magnetite nanoparticles have also been utilized for improvements in OLED properties. Gold-doped magnetite nanoparticles were demonstrated to efficiently enhance the luminescence of OLEDs due to the combination of light-scattering, plasmon resonance (gold), and magnetism (magnetite). Such a combination is very promising for the fabrication of new, highly efficient OLEDs.”. On the topic of magnetic field sensors (Page 20, Lines 622-627): “The main ideas of the abovementioned investigations were to combine different shapes of the optical fibers and optical grating with magnetic FFs to fabricate magnetic field sensors. The response of magnetic FFs to the magnetic field was used to measure the intensity of the external magnetic field. Such sensors are easy to fabricate, and they could find application in various fields, such as the power industry, optical sensing, and tunable photonic devices.”. On the topic of Magnetic Resonance Imaging (Page 26, Lines 808-815): “The investigations of positive (T1) non-toxic agents for MRI contrast enhancement as an alternative for gadolinium contrast agent as well as bimodal (FI/MRI) imaging agents have been widely developed. Exceedingly small superparamagnetic iron oxide NPs (up to ~10 nm) with various surface modifications (natural bovine serum albumin, artificial poly(acrylic acid)-poly(methacrylic acid, poly(allylamine hydrochloride), polyvinyl alcohol, zwitterion-coated, etc.) have been shown to have high T1 contrast power. Hybrid nanostructures consisting of carbon quantum dots and Fe3O4 nanoparticles demonstrating biocompatibility showed great potential as bimodal imagine agents.”. We added the table for comparison of mentioned applications (Page 28).
- “How this article stands out from previous review articles on this topic? An explanation is needed to defend the novelty of this review.”
Answer: We thank the reviewer for his comment and the explanation about the novelty of the review was added. The explanation for this point (Page 2, Lines 75-79): “This review not only promotes recent state-of-the-art methods of magnetite synthesis (biological routes) in comparison with routine chemical methods but also discusses novel cutting-edge applications of magnetite nanoparticles (magnetite ferrofluids) in optics and nanophotonics, such as “smart” windows, solar energy harvesting, organic light-emitting diodes, etc.”
- “The paper is not well-balanced. Almost half of the paper is written on the synthesis method and then other half on its applications in optics. Therefore, the title of the paper should be modified accordingly.”
Answer: We thank the reviewer for his comment and the title of the paper was modified accordingly. New title is “Magnetite nanoparticles: synthesis and applications in optics and nanophotonics”.
- “Author should make a comparison table to define which synthesis method is cost-effective, simple, and efficient.”
Answer: We thank the reviewer for his comment and apologize for the lack of clarity. Following the reviewer's comment, we added the table for comparison of synthesis methods and some recommendation concerning the choice of the synthesis method (Page 12-13).

Reviewer 4 Report
After careful evaluation I weel that revised version can be accepted.
Author Response
thank you very much!